# PERSONALIZED REPRESENTATION FROM PERSONALIZED GENERATION

**Shobhita Sundaram**[1*†]    **Julia Chae**[1*]    **Yonglong Tian**[2‡]    **Sara Beery**[1§]    **Phillip Isola**[1§]
[1]MIT    [2]OpenAI

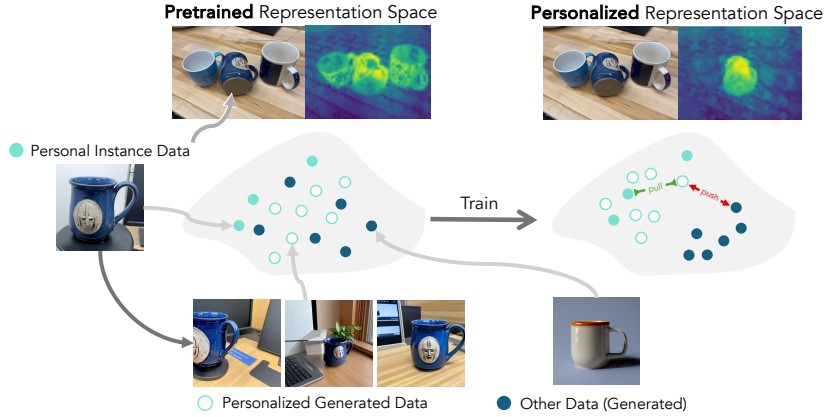

Figure 1: **Learning personalized representations from limited real data.** In this paper we explore whether and how synthetic data can be used to train a personalized representation. Given a few real images of an instance, we generate novel images and contrastively fine-tune a general-purpose pretrained model to learn a personalized representation, useful for diverse downstream tasks.

## ABSTRACT

Modern vision models excel at general purpose downstream tasks. It is unclear, however, how they may be used for personalized vision tasks, which are both fine-grained and data-scarce. Recent works have successfully applied synthetic data to general-purpose representation learning, while advances in Text-to-Image (T2I) diffusion models have enabled the generation of personalized images from just a few real examples. Here, we explore a potential connection between these ideas, and formalize the challenge of using personalized synthetic data to learn *personalized representations*, which encode knowledge about an object of interest and may be flexibly applied to any downstream task relating to the target object. We introduce an evaluation suite for this challenge, including reformulations of two existing datasets and a novel dataset explicitly constructed for this purpose, and propose a contrastive learning approach that makes creative use of image generators. We show that our method improves personalized representation learning for diverse downstream tasks, from recognition to segmentation, and analyze characteristics of image generation approaches that are key to this gain. Our project page: `https://personalized-rep.github.io/`

## 1 INTRODUCTION

Representation learning in computer vision seeks to learn general-purpose encodings for objects or semantic concepts that may be flexibly applied to downstream tasks such as recognition and semantic segmentation. In recent years we have seen a surge of interest in *personalized vision* – where a user

---

*Equal contribution.

†Work partially done as a student researcher at Google.

‡Work done at Google.

§Co-supervised, order by coin flip.

can easily develop customized models for objects of their personal interest, e.g., a model capable of detecting their pet dog in personally-collected images (Zhang et al., 2023; Cohen et al., 2022; Nitzan et al., 2022). Among other benefits, personalized systems can keep data private; preferably these models are trained locally, without needing to share user data to a centralized repository, or access other users' data. The personalized setting has two critical challenges. First, it is data-scarce: Curated data collection is time-consuming and expensive; a user would ideally need only provide a few examples of their object to obtain a personalized model. Second, it can be extremely fine-grained; e.g., recognizing an individual dog as opposed to recognizing the category "dog".

While modern vision models have proven successful for general-purpose tasks, adapting their representations to fine-grained problems with scarce labeled data remains challenging (Zhang et al., 2024; Radford et al., 2021; Cohen et al., 2022; Stevens et al., 2023). As shown in Figure 1, we contrast *general-purpose representations* with the notion of a *personalized representation*: a specialized representation space that encodes the knowledge about an instance of interest needed for a variety of downstream personalized tasks. In this paper, we ask: **Is it possible to learn a personalized representation from only a few real images of a single instance?**

Works such as (Tian et al., 2023b) have shown that, when intelligently paired with contrastive objectives, synthetic data can enable learning strong *general-purpose* visual representations. Other works have investigated *personalized generation* (Gal et al., 2023; Ruiz et al., 2022), but do not extend to representation learning. Our work targets the combination of these ideas: can personalized generation provide effective synthetic data for training *personalized representations*? We explore what makes for useful generative data augmentation for personalized representation learning and how to best learn from that data. We evaluate our learned representations for four downstream tasks: classification, retrieval, detection, and segmentation, and find that performance universally improves.

In summary, our contributions are the following:

- **Personalized representations** trained with synthetic data, using only three real examples of an instance, **significantly outperform pretrained counterparts** across datasets, backbones, and downstream tasks.

- We introduce new mechanisms for evaluating personalized representations, including **PODS – Personal Object Discrimination Suite – a new dataset** of 100 personal objects under specific distribution shifts, and reformulations of existing instance-level datasets.

- Leveraging **additional resources can significantly improve** personalized representations. While pretrained T2I models are key to achieving the best performance, comparable results can be obtained with **fewer computational resources**.

- Different generators introduce **unique biases/limitations** that affect representations.

## 2 RELATED WORKS

**Personalized visual generation.** Early efforts to personalize generated images attempted to edit specific people or styles given user inputs with Generative Adversarial Networks (GANs) (Bau et al., 2019; Roich et al., 2021; Alaluf et al., 2021; Dinh et al., 2022; Nitzan et al., 2022). Recent efforts focus on Text-to-Image (T2I) diffusion models, usually learning a unique identifier for a target object given a few images. Textual Inversion (Gal et al., 2023) freezes a pretrained generative model then learns a unique and personal text token for the object, which can be conditioned on for generation. NeTI (Alaluf et al., 2023) enhances expressivity and editability by learning different token embeddings for each diffusion timestep and U-Net layer. DreamBooth (Ruiz et al., 2022) fine-tunes the entire T2I model to produce more accurate images of the target concept. CustomDiffusion (Kumari et al., 2022) instead fine-tunes a subset of model weights, and enables joint training over multiple concepts. Follow-up works to these have sought to improve the efficiency and accuracy of personalized generations (Ruiz et al., 2023; Arar et al., 2023; Wei et al., 2023; Han et al., 2023; Guan et al., 2025), e.g., finetuning-free personalization methods that reduce computational cost (Shi et al., 2024; Chen et al., 2024; Huang et al., 2024; Ma et al., 2024).

**Personalized recognition and representations.** Personalized vision involves tailoring vision models to user-specific concepts and preferences. PerSAM (Zhang et al., 2023) extends the

Segment-Anything Model (Kirillov et al., 2023a) to segment user-specified objects with a few example images and masks. Personalization has also been explored for image captioning (Wang et al., 2023; Chunseong Park et al., 2017; Park et al., 2018), pose estimation (Nguyen et al., 2024b), and image retrieval via textual inversion: finding a mapping of images to text tokens (Saito et al., 2023; Karthik et al., 2023; Baldrati et al., 2023; Cohen et al., 2022; Yeh et al., 2023). Among the textual inversion works, PALAVRA (Cohen et al., 2022) enables personalization for both global and dense vision tasks but relies on large-scale captioned data for the inversion process. Our approach requires only a few images, without annotations, from the user. Concurrent works have also applied personalization to vision-language models for VQA and object recognition (Nguyen et al., 2024a; Alaluf et al., 2024). In contrast, we personalize general-purpose vision backbones using a self-supervised framework over generated data, achieving strong performance across both image-level and dense tasks without large-scale data. Recent concurrent work has also proposed PDM, which applies intermediate features of T2I models to personalized retrieval and segmentation Samuel et al. (2024).

**Re-Identification.** Personalized recognition is closely related to re-identification, in which a model is tasked with recognizing objects (Sun et al., 2004) or faces (Turk & Pentland, 1991) of the same identity. Early works in Re-ID explored metric learning on hand-crafted features (Ojala et al., 2002; Gray & Tao, 2008; Zhao et al., 2017); later methods learned deep metrics with supervised/unsupervised signals (He et al., 2021; Taigman et al., 2014; Schroff et al., 2015). Recent metric learning works use large curated datasets to train on thousands of unique instances of a certain category (typically humans (Zheng et al., 2015; Yadav & Vishwakarma, 2024) or vehicles (Liu et al., 2016; Amiri et al., 2024)). While our work involves training features with contrastive losses, we focus on personalizing pre-trained features for a single instance with a few images.

**Training on synthetic data.** Training on synthetic data has been extensively investigated to tackle issues like privacy preservation, data imbalance, and data scarcity (Sakshaug & Raghunathan, 2010; Tanaka & Aranha, 2019; Khan et al., 2019; Jahanian et al., 2021; Tucker et al., 2020). Diffusion models (Sohl-Dickstein et al., 2015; Ho et al., 2020) have further unleashed such potential in zero-shot settings (He et al., 2022b), few-shot settings (He et al., 2022b; Trabucco et al., 2023; Lin et al., 2023), out-of-distribution scenarios (Sariyildiz et al., 2023; Bansal & Grover, 2023; Jung et al., 2024), and supervised classification (Yeo et al., 2024; Kupyn & Rupprecht, 2024). These works note the importance of the classifier-free guidance scale (Sariyildiz et al., 2023; Tian et al., 2023b) and prompt selection (Lei et al., 2023), and propose post-processing filtering (He et al., 2022b) when using off-the-shelf T2I models. Alternatively, (Azizi et al., 2023) and (Yuan et al., 2023) fine-tune diffusion models on ImageNet and show improved classification performance when supplementing real with synthetic data. Similarly, (Zhou et al., 2023; Trabucco et al., 2023) invert training images as conditions for generating new synthetic images. Other studies address data-imbalance (Shin et al., 2023), domain shifts (Yuan et al., 2022), scaling synthetic data (Fan et al., 2023), and applications to various tasks, including segmentation (Wu et al., 2023), general-purpose representation learning (Tian et al., 2023b;a), and CLIP training (Hammoud et al., 2024).

## 3 METHODS

This paper tackles two questions: how to achieve personalized visual representation by leveraging generative models, and what factors are essential to producing highly effective training data. In Section 3.1 we formalize the personalized representation task. Our three-stage method is then illustrated in Figure 2. We prepare a personalized generator from a few target instance images (Section 3.2) and produce synthetic personalized data (Section 3.3). We then train a personalized representation on the generated data with a contrastive objective (Section 3.4). Lastly, we consider scenarios with additional annotations and data, and how to incorporate them to enhance personalization.

### 3.1 FORMALIZING THE PERSONALIZED REPRESENTATION CHALLENGE.

We assume access to a small dataset of real images $\mathcal{D}_R$ of a specific object $c$, and the generic category $c_{pr}$ of the object. We use a generative model $g_\theta(x)$ to synthesize a novel dataset $\mathcal{D}_S$ of images of $c$ and train a personalized representation by adapting a general purpose vision encoder $f_\phi$.

We assume we are only provided images of $c$ (which we also denote as an *instance*) for training our personalized representation. We evaluate on global and local downstream tasks. Note that we

evaluate *instance* performance (e.g., one v. all classification, detection, etc). This differs from many previous works that focus on generating synthetic data for closed-set $k$-way classification (Shin et al., 2023; He et al., 2022b).

## 3.2 PERSONALIZED SYNTHETIC DATA GENERATION

We generate personalized data from $\mathcal{D}_R$ using Stable Diffusion 1.5, a T2I model, as our generator $g_\theta$. We adapt $g_\theta$ using DreamBooth (Ruiz et al., 2022) to generate novel images of $c$ when conditioned on an identifier token.

A T2I diffusion model $g_\theta$ generates images given an initial noise latent $\epsilon \sim \mathcal{N}(0,1)$ and a conditioning text embedding $\hat{y} = \Gamma_\omega(y)$ where $\Gamma_\omega$ is a text encoder, and $y$ is a user-provided prompt. Given a ground-truth image $x$ and the text embedding $\hat{c_{pr}}$ of the generic semantic category $c_{pr}$, DreamBooth fine-tunes $g_\theta$ using the loss:

$$\mathbb{E}_{x,\hat{y},\epsilon,\epsilon',t}[w_t||g_\theta(\alpha_t x + \sigma_t \epsilon, \hat{y}) - x||_2^2]$$
$$+ \lambda w_{t'}||g_\theta(\alpha_{t'} x_{pr} + \sigma_{t'}\epsilon', \hat{c_{pr}}) - x_{pr}||_2^2,$$

where $x_{pr}$ is an image synthesized with the pre-trained generator conditioned on $\hat{c_{pr}}$, $t$ is the timestep, and variables $\alpha_t$, $\sigma_t$, and $w_t$ relate to the noise schedule and sampling quality. The first loss term is a reconstruction loss on $x$, and the second term is a prior preservation loss on $x_{pr}$. The two loss terms are weighted by $\lambda$. Following standard implementations, we also fine-tune $\Gamma_\omega$ with the same loss. For further details, refer to (Ruiz et al., 2022).

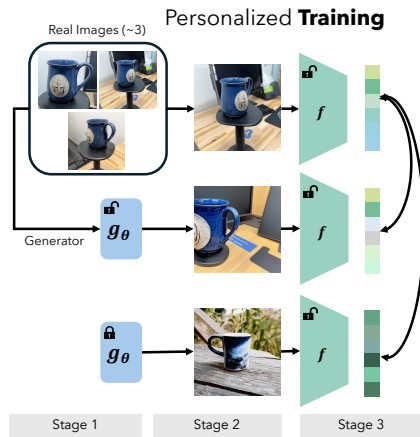

Figure 2: **Personalized Representation Training Pipeline.** Our three-stage training method: 1) Generative Model Training 2) Synthetic Data Generation 3) Contrastive LoRA Fine-Tuning.

While there are several alternative methods for personalized generation (Gal et al., 2023; Alaluf et al., 2023), we focus on DreamBooth, which has been shown to maintain highest fidelity to fine details (Alaluf et al., 2023).

## 3.3 CONTROLLING GENERATED DATASET ATTRIBUTES

Prior work has observed that fidelity to the target subject and diversity of generated data are both important factors (Sariyildiz et al., 2023). T2I models offer several mechanisms of injecting diversity into generated outputs, allowing us to explore the relationship between these attributes and the quality of learned personalized representations.

**Classifier-Free Guidance (CFG).** A common way of injecting diversity for diffusion models is modifying the CFG (Ho & Salimans, 2022) at inference, which controls how strongly the generation adheres to the conditioning prompt. We experiment with CFG $\in \{4.0, 5.0, 7.5\}$.

**LLM-generated captions.** As seen in (He et al., 2022b) and (Dunlap et al., 2023), off-the-shelf Large Language Models such as T5 (Raffel et al., 2023) can be leveraged to generate text-prompts for each object. Following prior works, we generate image captions with GPT-4 (OpenAI, 2023), ensuring that they introduce rich context descriptions in addition to describing the target object. For example, if the object is a shirt, an LLM-generated prompt could be `"a shirt on a coat hook"`, or `"a person wearing a shirt at a street market"`. Details in A.4.1.

## 3.4 REPRESENTATION LEARNING FROM SYNTHETIC DATA.

Given $(\mathcal{D}_R, \mathcal{D}_S)$ of instance $c$, we personalize $f_\phi$ via fine-tuning. Critical to representation learning is having both positive and negative examples. We obtain positives from $\mathcal{D}_S$. We generate negatives $\tilde{\mathcal{D}}_S$ by prompting the pretrained $g_\theta$ (Stable Diffusion 1.5) with the generic object category: `"a photo of $c_{pr}$"`.

Given $(x, x_+, x_0, ..., x_N)$ where $x \in \mathcal{D}_\mathcal{R}$, $x_+ \in \mathcal{D}_\mathcal{S}$, $x_i \in \tilde{\mathcal{D}}_\mathcal{S}$ for $i = 0, \ldots, N$, we extract $f_\phi$ features as a concatenation of the `CLS` token and average-pooled final-layer patch-embeddings. We

then finetune $f_\phi$ using the infoNCE loss,

$$\mathcal{L}_{\text{InfoNCE}} = -\log \frac{\exp(\text{sim}(\mathbf{x}_0, \mathbf{x}_+)/\tau)}{\sum_{i=1}^{N} \exp(\text{sim}(\mathbf{x}_0, \mathbf{x}_i)/\tau)}.$$

This loss pushes together the representations of real and synthetic images of $c$, and pushes apart representations of $c$ and other instances. We also experiment with alternate contrastive/non-contrastive losses in the Appendix (C.1). We fine-tune via Low-Rank Adaptation (LoRA), which is more parameter-efficient than full fine-tuning (Hu et al., 2021).

### 3.5 Alternatives to DreamBooth

Given the computational cost of DreamBooth, we also explore alternative datasets:

**Real data baseline.** A simple baseline is to contrastively fine-tune using only the available real data $\mathcal{D}_R$ as positives. Here, we still use a large pool of negatives.

**Comparisons enabled by extra resources.** With so few real images, there may be benefit in expending effort upfront to collect further labels and data. A user might annotate their images, download internet-available data, or even capture more images of the target object. We describe possible cases below.

*Segmentation masks:* We consider collecting segmentation masks of $\mathcal{D}_R$. This allows for a simple, cheap generative model: *Cut-and-Paste*. Here the generator samples independently from foregrounds containing the target object (carved from $\mathcal{D}_R$) and generic backgrounds from a T2I model (details in A.4.3). With masks, we can also improve DreamBooth generations by enabling *masked DreamBooth training* and *filtering*. Fine-tuning $g_\theta$ can be affected by signals such as shared backgrounds. To minimize such overfitting, we mask out the gradients for background pixels during DreamBooth training, as in (Zhang et al., 2023). This leads to more diverse generations with better prompt adherence. We also use masks to filter generated datasets. Using a perceptual metric (Fu et al., 2023) and perSAM Zhang et al. (2023) we predict a mask for the generated image and measure the similarity to masked training images, filtering out those below a threshold. Details in A.4.2.

*Internet-available real data:* A user may download open-source real datasets; these can provide a source of *real negatives* and *real backgrounds* for Cut/Paste, avoiding the computational cost of image generation and enabling comparison to real-only approaches.

*Extra real positives:* A user may expand $\mathcal{D}_R$ by physically collecting extra real target images. This also provides an expanded set of images for Cut/Paste and DreamBooth generation.

**Experimental analysis.** In Section 5.2 we explore if there are still benefits to using generated data versus computationally cheaper alternatives if these extra annotations or real data are available.

## 4 Experiments

### 4.1 Datasets

Evaluating our personalized representations necessitates instance-level datasets with multiple tasks, across various real-world scenarios. To satisfy this criteria, we reformulate two existing datasets – DeepFashion2 (Ge et al., 2019) (focused on shirts) and DogFaceNet (Mougeot et al., 2019) (focused on dogs) – and introduce a new dataset, PODS (Personal Object Discrimination Suite). PODS features common personal and household objects, enabling instance-level evaluation across classification, retrieval, detection, and segmentation tasks. To assess robustness and generalization, DF2 and Dogs provide in-the-wild test images, and DF2 and PODS include test sets designed with distribution shifts. All datasets are split such that for each object there are exactly 3 training images and at least 3 test images. We summarize our datasets and procedures below; for additional details and sample images, refer to Section A of the Appendix.

**DeepFashion2 (DF2)** is a large-scale fashion dataset with 873K Commercial-Consumer clothes pairs for instance-level retrieval, detection, and segmentation. We use the Consumer-to-Shop Clothes retrieval benchmark, which matches gallery images of clothing items to in-the-wild consumer images, thus encoding a train-test distribution shift. Out of 13 clothing categories, we select

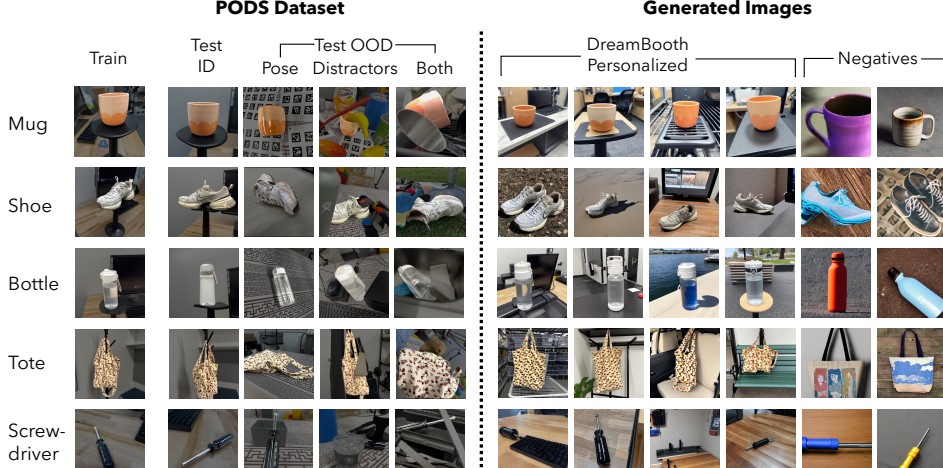

Figure 3: **(left) Examples of instances from our new PODS dataset.** We showcase one example instance from each of the five object categories, displaying images from both the training and various test splits. We dim the surrounding scene, highlighting the instance of interest. This masking technique is not applied to our dataset images or during training. **(right) We show example generated images from Dreambooth** (LLM, cfg 5), which we use as positives in our representation learning finetuning.

the *shirts* category as our focus. We subselect a set of 169 shirts, after filtering out categories which lack sufficient numbers of gallery images.

**DogFaceNet (Dogs)** is a dog identification dataset, containing 8600 images of 209 dogs. Dog-FaceNet includes multiple unique dogs of the same breed, making the dataset more challenging. We subselect 80 dogs with sufficient numbers of images, and split the images into a train and test set. To support evaluation of segmentation and detection, we manually annotate the dataset with masks.

**Our new dataset: PODS** contains 100 unique objects across 5 every-day categories (mugs, screw-drivers, shoes, bags, waterbottles). Each object is captured in four scenes with varying conditions and vantage points. The train set contains 3 images of each object, displayed in a canonical pose with full visibility of key identifying features such as logos. The test set contains 80-100 images of each object, captured in four scenes: one in-distribution (ID) and three out-of-distribution (OOD). We show examples of each type of scene in Figure 3. The ID scene is taken in the same conditions as the training images. OOD scenes include one scene with *pose variation*, one scene with *distractor objects*, and one scene with *both variations*. All OOD scenes are against differing backgrounds from the ID scenes.

The dataset supports evaluation across 4 tasks: classification, retrieval, detection, segmentation. Each test image is associated with the target instance label. From each test scene, 3 randomly-selected images are additionally annotated with the bounding box and segmentation mask of the displayed object. Masks are manually annotated using TORAS (Kar et al., 2021) and SAM (Kirillov et al., 2023b); detection bounding boxes are extracted from the masks. We expect the PODS dataset to be a meaningful benchmark for personalized representation and instance-level detection research, and a valuable resource for the personalized generation community.

## 4.2 TRAINING

We fine-tune a vision backbone $f_\phi$ on sets of $(x, x_+, x_0, ...x_N)$ where the anchor $x$ is drawn from the 3 real positive images, the positive $x_+$ is drawn from the pool of synthetic positives, and $x_i$ for $i = 0, \ldots, N$ are drawn from the synthetic negatives. We apply the following data augmentations to all images: random rotations, horizontal flips, and resized crops. We experiment with state-of-the-art backbones: DINOv2-ViT B/14 (Oquab et al., 2023), CLIP-ViT B/16, (Radford et al., 2021), and MAE-ViT B/16 (He et al., 2022a).

Each dataset is randomly divided class-wise into a validation set (30 classes), and test set (varying size). We train one model per class and report the mean performance over test classes. Using the

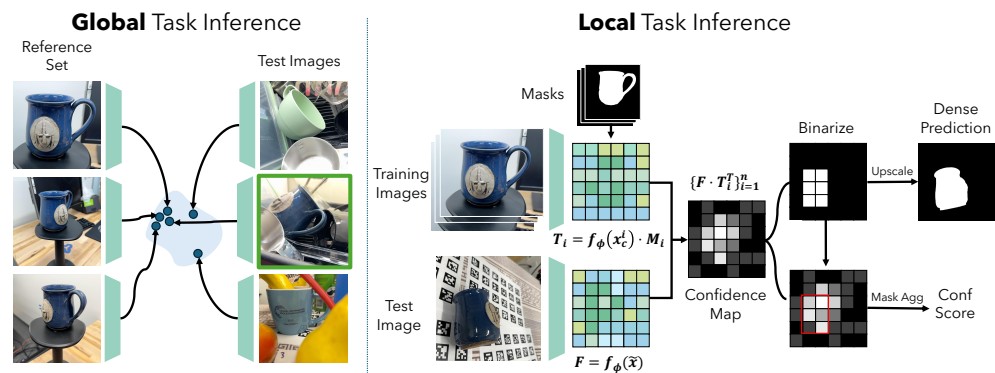

Figure 4: **Inference Pipelines**. We visualize the global (classification, retrieval) and local (detection, segmentation) evaluation pipelines. Global inference uses cosine similarity between CLS embeddings, while local inference extracts patch features with spatial information.

validation set we sweep over key training parameters: # synthetic positives, # anchor-positive pairs, and the choice of loss function. Based on this sweep we LoRA finetune with the infoNCE loss for 2 epochs over 4500 anchor-positive pairs, drawn from 450 synthetic positives and 1000 synthetic negatives. For validation results and training details, refer to the Appendix (B.1, C.1 ).

### 4.3 EVALUATION

We evaluate personalized representations across one v. all tasks that require *global understanding*, and *the ability to localize* with respect to the target object. Due to the few-shot nature of our task, we evaluate representations directly, without training task-specific heads. We summarize our evaluations below and in Figure 4.

**Classification.** For a particular instance $c$, given a test image $\tilde{x}$, a frozen encoder $f_\phi$, and training images $x_i^c \in \mathcal{D}_R$ we compute the maximum cosine similarity between the CLS tokens of $f_\phi(\tilde{x})$ and $f_\phi(x_c^i)$; this is taken as the prediction confidence. Samples with confidence above some threshold $t$ are taken as positives. Thus we report the Area under the Precision-Recall Curve (PR-AUC), which is a threshold-free metric.

**Retrieval.** We use our test set as the "query" set, and $\mathcal{D}_R$ as the "retrieval" set. We compute the cosine similarity between the CLS token of $f_\phi(\tilde{x})$ and those of the images in $\mathcal{D}_R$. We score the resultant ranking with the NDCG metric (Jeunen et al., 2024).

**Segmentation.** We compute the average cosine similarity between the patch embeddings of $f_\phi(\tilde{x})$ and those of $f_\phi(x_c^i)$ to generate a local confidence map, where $x_c^i$ are masked to the target, following the procedure of (Zhang et al., 2023). We then apply binarization directly to the confidence map using Otsu's thresholding method (Otsu et al., 1975) and upscale to the image dimensions to yield a segmentation prediction. We report the standard mask AP metric (Deng et al., 2024) and F1 scores, given the high imbalance between positives and negatives in the test sets.

**Detection.** We apply the same procedure as segmentation, and extract a bounding box prediction by drawing a box around the predicted mask. We obtain a confidence score for the box by averaging over the confidence map within the box region. We report the standard AP metric and the F1 score.

For each task, we compare the performance of our learned personalized representations to pretrained models. Note that we do not train prediction heads, due to the lack of real training data in our setting – rather, we use these evaluations to probe what our personalized features learn about the target object, compared to pretrained features.

## 5 RESULTS AND DISCUSSION

### 5.1 PERSONALIZED REPRESENTATIONS IMPROVE OVER PRETRAINED REPRESENTATIONS

We LoRA-tune three backbones (DINOv2, CLIP, MAE) and evaluate the personalized representations on four tasks. We sweep over synthetic datasets with different levels of diversity by varying the CFG and usage of LLM-generated prompts as described in Section 3.3, and select the best for each

| | Classification | | | Retrieval | | | Detection | | | Segmentation | | |
|---|---|---|---|---|---|---|---|---|---|---|---|---|
| | PODS | DF2 | Dogs | PODS | DF2 | Dogs | PODS | DF2 | Dogs | PODS | DF2 | Dogs |
| DINOv2 | 28.1 | 14.4 | 83.1 | 69.6 | 36.3 | 89.4 | 11.0 | 5.3 | 12.2 | 12.9 | 4.7 | 12.5 |
| DINOv2-P (DB) | 48.0 ↑ | 35.7 ↑ | 81.6 ↓ | 79.6 ↑ | 64.1 ↑ | 94.6 ↑ | 12.6 ↑ | 9.8 ↑ | 17.4 ↑ | 13.8 ↑ | 9.3 ↑ | 17.2 ↑ |
| CLIP | 26.7 | 12.7 | 36.4 | 61.4 | 34.7 | 58.0 | 0.1 | 2.9 | 6.1 | 0.3 | 2.9 | 7.3 |
| CLIP-P (DB) | 47.4 ↑ | 26.7 ↑ | 65.0 ↑ | 71.7 ↑ | 51.0 ↑ | 80.9 ↑ | 0.8 ↑ | 4.8 ↑ | 9.5 ↑ | 1.6 ↑ | 4.9 ↑ | 10.9 ↑ |
| MAE | 8.7 | 5.2 | 11.3 | 34.6 | 25.8 | 33.6 | 0.2 | 1.4 | 1.4 | 0.4 | 1.2 | 0.7 |
| MAE-P (DB) | 17.0 ↑ | 12.2 ↑ | 29.9 ↑ | 30.7 ↓ | 23.7 ↓ | 42.7 ↑ | 0.4 ↑ | 1.5 ↑ | 1.5↑ | 0.5 ↑ | 1.2 = | 0.8 ↑ |

Table 1: **Performance of personalized v. pretrained representations across backbones, tasks, and datasets.** We compare personalized and pretrained backbones, assuming access to only 3 real images and a T2I model with no extra data/annotations. For each backbone we report results for the best-performing synthetic dataset (chosen using the validation set), averaged over 3 seeds. Personalized representations (-P) largely outperform pretrained representations across all tasks. Full results across synthetic datasets, and error over seeds, are in the Appendix (C.2).

| Method | Real Backgrounds | Real Negs | Classification | | | Retrieval | | | Detection | | | Segmentation | | |
|---|---|---|---|---|---|---|---|---|---|---|---|---|---|---|
| | | | PODS | DF2 | Dogs | PODS | DF2 | Dogs | PODS | DF2 | Dogs | PODS | DF2 | Dogs |
| Real Imgs | - | ✗ | 32.4 | 28.5 | 81.5 | 59.3 | 51.7 | 92.6 | 10.8 | 8.0 | 14.7 | 12.0 | 6.9 | 15.0 |
| | - | ✓ | 34.4 | 28.5 | 81.5 | 60.6 | 51.7 | 92.5 | 11.7 | 8.0 | 14.7 | 13.2 | 6.9 | 15.0 |
| Cut/Paste | ✗ | ✗ | 60.4 | 48.5 | 83.3 | _82.0_ | 68.3 | 93.7 | 14.6 | 13.1 | 16.2 | 18.4 | 11.9 | 16.2 |
| | ✓ | ✓ | 58.7 | 47.3 | _87.8_ | 77.6 | 66.0 | _95.8_ | 15.3 | 12.9 | 13.0 | 19.8 | 11.2 | 14.0 |
| Masked DB | - | ✗ | 58.1 | 47.8 | 83.1 | 81.0 | 69.3 | 94.1 | _16.3_ | 14.0 | **17.3** | 19.3 | 12.5 | _17.2_ |
| | - | ✓ | 55.6 | 43.3 | 84.8 | 76.3 | 68.4 | 94.1 | 16.2 | 13.7 | 12.7 | _19.9_ | 11.2 | 13.6 |
| Combined | ✗ | ✗ | **61.9** | **51.6** | 84.5 | **83.9** | **71.3** | 94.6 | 14.8 | **14.8** | **17.3** | 19.4 | **13.6** | **17.3** |
| | ✓ | ✓ | _61.6_ | _49.6_ | **88.4** | 80.2 | _70.4_ | **96.2** | **16.9** | _14.6_ | 13.4 | **21.5** | _12.7_ | 14.6 |

Table 2: **Comparisons across data augmentation methods.** We compare DINOv2-P trained with different augmentation strategies, including those requiring extra annotations/data. Training with synthetic data improves performance significantly over training with the limited real-image dataset; combined Masked DreamBooth + Cut/Paste performs best in all cases. Significant boosts are also achievable more cheaply when incorporating internet-available real data with Cut/Paste.

backbone based on validation performance. In Table 1 we compare the performance of pretrained and personalized backbones on DF2, Dogs, and PODS, using the best synthetic dataset for each backbone. Full results are in the Appendix (C.2).

Personalized models (-P in Table 1) boost performance in 32/36 cases. We observe improvements – often substantial – in nearly every combination of backbone and task besides MAE retrieval. For example, averaged across datasets, DINOv2 detection improves by $48\%$, DINOv2 retrieval by $32\%$, and CLIP classification by $89\%$ relative to pretrained models. Across all three datasets, personalized models boost performance both for global tasks requiring semantic understanding, and dense tasks requiring localization. We visualize dense prediction maps for multi-object images in Figure 5 and the Appendix (D.1), showing that personalized patch features better localize the target object. We also visualize challenging success and failure cases in the Appendix (D.2).

## 5.2 What makes for the best training data?

In the previous section, we show that personalizing representations significantly boosts personalized task performance. Here, we compare various data generation approaches (Section 3.5), some leveraging additional resources, to investigate tradeoffs between computational cost and performance.

Results are shown in Table 2, with the runtime for each generation method in Table 3 (Appendix A.4.4). We first explore personalizing models only with the existing real training images in $\mathcal{D}_R$ (Real Imgs). We then examine incorporating segmentation masks of $\mathcal{D}_R$. This enables a cheap baseline method, Cut-and-Paste (CP), and improvements to DreamBooth via masked training and filtering (Masked DB). We also test sampling from a combined pool of CP and Masked DB images

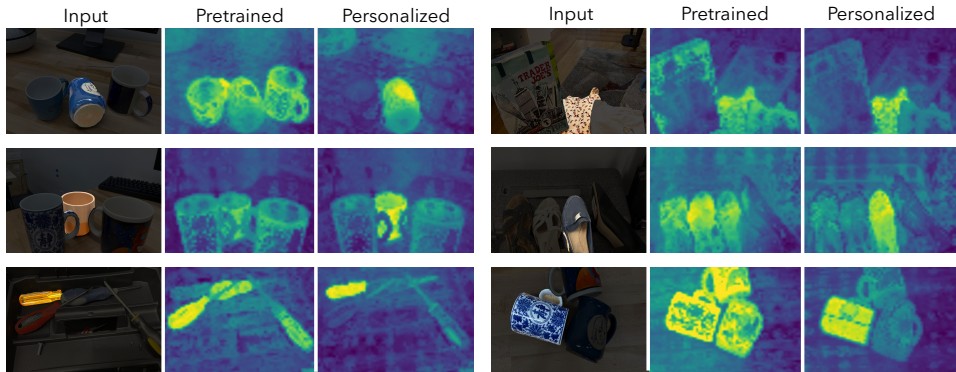

Figure 5: **Qualitative Results.** Each triplet shows the test image (left), dense prediction maps for pretrained DINOv2 (center), and personalized (right). Prediction maps are computed via patchwise embedding similarity between the test and localized train images following Figure 4. Personalized representations distinctly localize the target instance, unlike pretrained embeddings. For visualization only, the personalized instance is highlighted in the test images but this is not applied during training or inference.

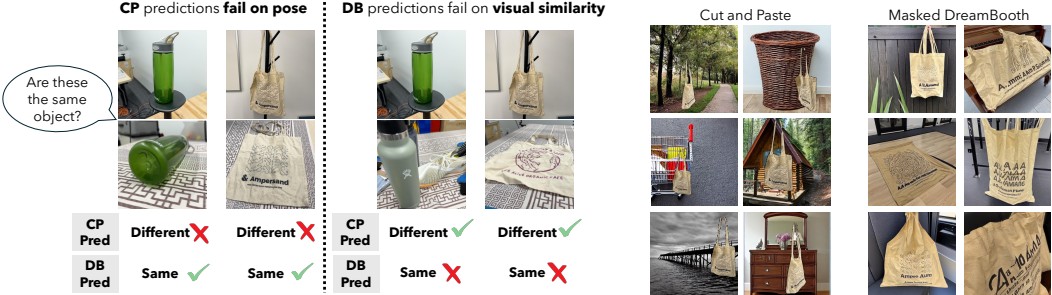

Figure 7: **DreamBooth vs Cut and Paste Model Failures.** We show object pairs where DB-personalized and CP-personalized models differ most in predictions.

Figure 8: **(left)** CP limitations include pose and realism. **(right)** Masked DB struggles with fine-grained details.

(Combined). All synthetic data pools are each 450 images for fair comparison. We also ablate the use of generated negatives and generated CP backgrounds by using open-source images as real alternatives; this enables comparison to cheap methods that use only real images.

Synthetic data methods outperform real-image-only methods, with the Combined pool performing best across all datasets/tasks. Performance improves significantly from incorporating masks; results in Table 2 out-perform DINOv2-P results in Table 1, obtained without masks. Incorporating real negatives and real Cut/Paste backgrounds does not consistently improve performance over their generative counterparts. Comparing *performance* (Table 2) and *runtime* (Table 3) reveals a tradeoff between the two. The high performance of Combined indicates that learned models provide valuable knowledge. However, sampling CP with real backgrounds performs similarly to Masked DB alone, showing that strong performance can still be achieved with an efficient alternative.

**Scaling real positives.** We collect extra training images for 25 PODS instances (5 per category) with different backgrounds, poses, and lighting. In Figure 6 we compare training DINOv2-P on $\mathcal{D}_R$, and on synthetic $\mathcal{D}_S$ generated from $\mathcal{D}_R$ using the Combined method. Performance increases as $|\mathcal{D}_R|$ increases, saturating at $|\mathcal{D}_R| = 15$, likely due to limitations in the diversity of the additional real data. Expanding diversity further could improve scaling but requires significant manual effort. Synthetic augmentation remains effective as $|\mathcal{D}_R|$ scales (27% gain with 3 real images, 8% gain with 20). As generative models improve, the ability to complement real datasets is expected to grow.

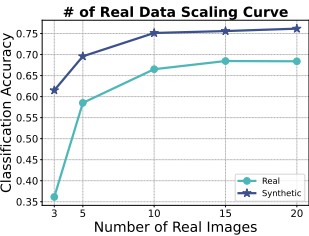

Figure 6: **Real and synthetic data scaling curve for a subset of PODS.**

### 5.3 How do different datasets affect representations?

As seen in Table 2, Masked DB and CP achieve similar performance. Here, we show that they exhibit distinct strengths and limitations. We analyze divergence cases in high-confidence predictions of DB- and CP-trained representations, revealing consistent failure patterns. DB-trained models excel at pose generalization but often confuse visually similar instances. Conversely, CP models are more robust to distractors but falter when encountering unfamiliar poses. We show examples in Figure 7. This trend is also quantitatively shown in Figure 9. In the PODS Distractors split, CP models outperform DB models by 7%, whereas in the Pose split, DB models surpass CP models by 6.4%.

In Figure 8 we trace these attributes of learned representations to biases/limitations in DB and CP generations. The main DB limitation is difficulty in preserving fine-grained object details, resulting in images that only loosely resemble the target characteristics, even with filtering. These inaccuracies likely propagate to the learned representation, compromising its fine-grained discriminative ability. Conversely, CP maintains perfect object fidelity but restricts pose variability, potentially leading to pose overfitting.

We also study the impact of fidelity and diversity in Figure 12 (Appendix C.4). Without filtering, DB datasets lie at extremes (high-diversity, low-fidelity or vice versa). Both the Masked DB and CP datasets achieve a better balance, and thus higher performance.

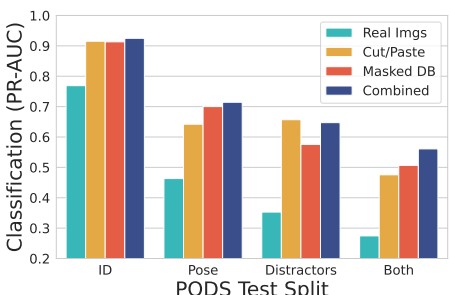

### 5.4 Applications

Our thresholding evaluation for dense tasks allows direct probing of personalized patch features, however does not achieve state-of-the-art results. We show that personalized representations can be easily integrated into perSAM – a practical pipeline – to improve its performance (Zhang et al., 2023). Instead of extracting keypoint proposals and confidence scores with pretrained DINOv2, we use DINOv2-P. Segmentation performance improves from 5.8% F1 score to 10.9 % on DF2, 20.9% to 26.2% on Dogs, and 20.8% to 24.7% on PODS. Full details/results are in the Appendix (C.3).

Figure 9: **DB, CP, and Combined performance on PODS test splits.** DB outperforms CP on the Pose split, and underperforms it on the Distractors split, whereas Combined performs best on both. These results support our qualitative observations.

## 6 Limitations and Conclusions

### 6.1 Limitations

Our pipeline has potential for high computational cost (Table 3) from T2I models. We defer this limitation to future research in T2I efficiency. A cost-conscious user may choose instead to use a cheaper method such as Cut/Paste with worse/comparable performance. There are also cases where the performance boost from T2I models may be worth the cost (e.g., personalized medicine, re-ID for rare animals, identifying items for low-vision people (Morrison et al., 2023)). Furthermore, by training on data generated by T2I models we may inherit their biases and limitations (see 5.3).

### 6.2 Conclusions

We leverage generative models to adapt general-purpose representation spaces to personalized ones, using *very few* real examples of a *single* instance. We quantitatively and qualitatively show that personalized representations consistently boost performance across downstream tasks. Moreover, we also study computationally cheaper alternatives that leverage additional resources, and show that combining different generation methods may enable further improvements. We release our new dataset, PODS, and new splits and annotations for two existing instance-level datasets, offering comprehensive benchmarks for future work.

In the future, generative models will continue to get faster, cheaper, more accurate, controllable and potentially less biased. Our work is not limited to existing generation techniques. We are excited by the potential of learning personalized representations in this way, and envision a possibility that allows users to have ownership over their own models and data.

CODE OF ETHICS

The authors have read the code of ethics and we acknowledge that we adhere to the code presented.

ETHICS STATEMENT

Below, we discuss potential societal impacts of research in personalized representations.

One potential application of such an approach, and more broadly of all instance-level recognition and re-identification work, is surveillance technology, which may infringe on human privacy depending on the use-case and intent. In this paper we focus on non-human (object-centric) applications and datasets, as opposed to facial or vehicle re-identification.

By removing the need for real negatives, our approach potentially enables personalized models that do not require people to share their data in a central repository or access others' data during personalization, beyond the prior contained in a pretrained T2I model. Users may choose to keep personal data siloed (similarly to federated learning settings), while still benefiting from personalized models. Removing the need for external data reduces data storage/access requirements for users.

Computational cost (discussed in Section 6.1) adds another important dimension: The power and water usage by the GPUs needed for fine-tuning and T2I inference. AI has become an increasingly large portion of global power use, water use, and carbon emissions Luccioni et al. (2023); inefficient methods exacerbate this effect. Users prioritizing cost over optimal performance may choose computationally cheaper alternatives that do not require T2I models (Section A.4.4).

Our work includes reformulations of two existing datasets and a novel dataset. Privacy and consent measures for all three datasets are detailed in Section A.3, ensuring transparency and ethical compliance.

REPRODUCIBILITY

Our website (https://personalized-rep.github.io/) and Github repository (https://github.com/ssundaram21/personalized-rep) contain the source code for our work, including the necessary metadata to reproduce results, such as the LLM-generated captions used for dataset synthesis. The codebase features end-to-end pipelines for data generation, personalized representation training, and inference/evaluation. In the appendix, we provide detailed hyperparameters and outlines for data generation (DreamBooth finetuning, Cut and Paste), representation training (parameters for LoRA and other model training hyperparameters), and our evaluation pipelines. We also present extensive ablations and full results in the appendix for transparency, justifying each design choice so that other researchers can replicate our findings. Finally, we release our dataset, PODS, and the reformulated DeepFashion2 and DogFaceNet datasets.

ACKNOWLEDGMENTS

This material is based on work that is partially funded by an unrestricted gift from Google through the MIT-Google Program for Computing Innovation, and by the Defense Science and Technology Agency, Singapore. This work was also supported by the AI and Biodiversity Change (ABC) Global Climate Center, which is funded by the US National Science Foundation under Award No. 2330423 and Natural Sciences and Engineering Research Council of Canada under Award No. 585136. S.S. is supported by an NSF GRFP fellowship. J.C. is supported by an NSERC PGS-D fellowship.

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

# A  DATASETS

## A.1  DF2 AND DOGS REFORMULATION

We reformulate two datasets: DeepFashion2 and DogFaceNet.

**DeepFashion2.**  In this section, we identify the training split that we subselected from DeepFashion2. DeepFashion2 is a large-scale retrieval dataset with a public train-validation-test split. Each split contains its own query/customer and gallery set. We sample our training set from the gallery set of the validation split, and we sample our test set from the consumer set of the validation split. Each image in validation split has a six-digit identifier and an annotation file containing the information. We organize the data into instance categories; we define the training and testing images for an instance as gallery/consumer images depicting the same clothing item of the same style.

Below we provide metadata for our subselected dataset.

- **Clothing Item Category:** Short-Sleeve Tops, category id 1
- **Unique Instances Selected:** 169
- **Total # Training Images:** 507
- **# of Training Images per Instance:** 3
- **Total # Test Images:** 1271
- **Range of Test Images per Instance:** [4, 24]

**Dogs.**  Our Dogs dataset reformulates the DogFaceNet_large split from the datasets released with DogFaceNet for dog re-identification studies. Since the original dataset was published with instance-level splits, we perform our own splitting of the dataset to fit our personalized learning setting. Due to the nature of data collection (images of dogs collected from sequential footage), we had to pay careful attention to the possibility of data poisoning between the train and test set. The procedure we followed for data splitting is as follows:

1. Filtered DogFaceNet dog classes to keep classes with above 10 images per instance
2. Performed a random train-test split for every instance, keeping 3 images for train and rest for test
3. Manually inspected every instance in the dataset to remove data poisoning. This entailed looking through the training and test data and making sure that no test images were from the same sequential footage as the train data. When such images were discovered, they were removed from the test set.
4. After data-cleanup, we removed instances with less then 4 remaining test images.

The above procedure resulted in 80 total dog classes. Below is the metadata of our subselected dataset.

- **Unique Instances Selected:** 80
- **Total # Training Images:** 240
- **# of Training Images per Instance:** 3
- **Total # Test Images:** 1218
- **Range of Test Images per Instance:** [6, 38]

**Dataset Examples.**  In Fig. 10 and 11 we show examples of training and test images for the Dogs and DF2 datasets. Notably, the test sets cover a wide range of diverse in-the-wild scenarios, including different contexts, positions, backgrounds, camera angles, occlusions, and lightings.

**Licenses for existing assets**  The datasets we use are released under the following licenses:

DeepFashion2: MIT
DogFaceNet Dataset: Non-Commercial, Research-Only

**DF2 Dataset**

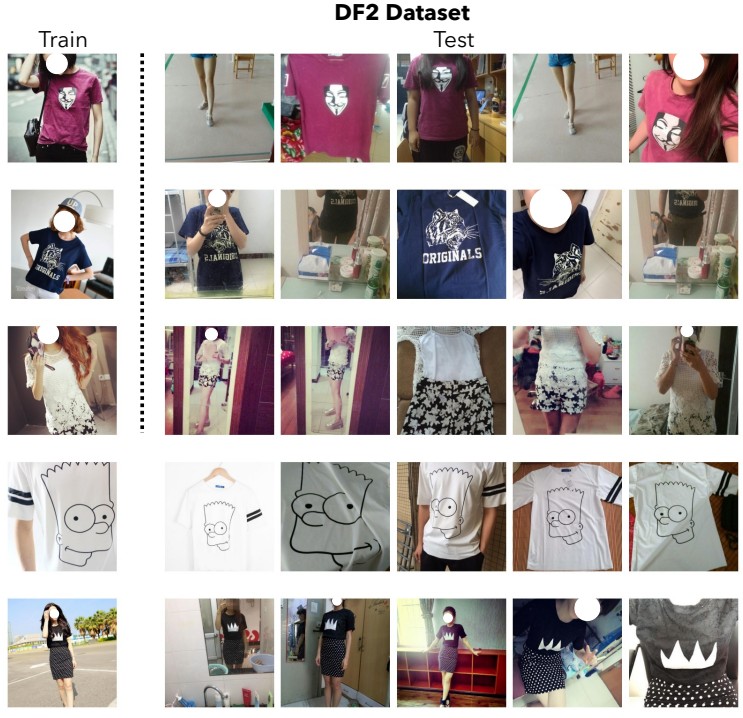

Figure 10: **DF2 train/test examples.** Training images are of models wearing clothes, and test images are from consumers. Images and classes are randomly sampled.

**DOGS Dataset**

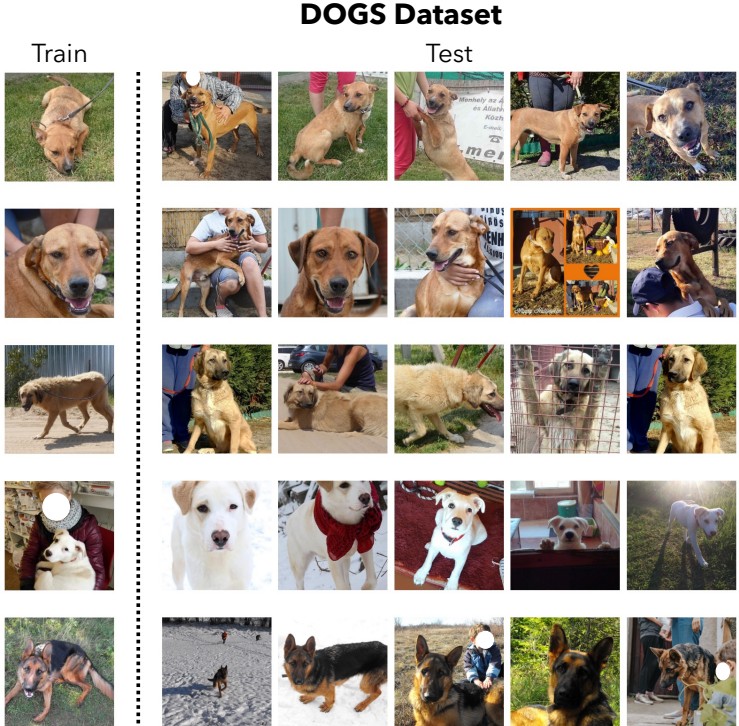

Figure 11: **Dogs train/test images.** Images and classes are randomly sampled.

## A.2 PODS

### A.2.1 PODS OVERVIEW

- **Unique Instances:** 100
- **Total # Training Images:** 300
- **# of Training Images per Instance:** 3
- **Total # Test Images:** 10888
- **Range of Test Images per Instance:** [71, 201]
- **Total # Test Images with Dense Annotations:** 1200
- **Range of Test Images per Instance with Dense Annotations:** [12, 12]

### A.2.2 PODS CREATION

PODS includes images of 100 objects; 20 each from five categories: mugs, screwdrivers, tote bags, shoes, and water bottles. We choose these categories to cover a range of personal, everyday objects, both rigid and deformable.

**Scenes.** Every object is captured in 4 scenes. We describe each scene through 4 attributes: the *background* of the scene, the *pose* of the target object, the presence of *distractor objects*, and the visibility of the object's *key identifying features* (such as a mug's logo).

Below is a detailed description of each scene.

- **Train/In-distribution:**
    - *Background:* The object is against a plain office background.
    - *Pose*: The object is located on a flat surface, upright in its "canonical pose".
    - *Distractors:* No similarly-sized distractor objects are nearby, or in the clear foreground.
    - *Identifying features:* Key identifying features (i.e. logo, handles, etc) clearly visible.
- **Distractors:**
    - *Background:* The object is against different background from the Train scene.
    - *Pose*: Object is upright in its "canonical" pose.
    - *Distractors:* Object is surrounded by 2-5 distractor objects of varying sizes, located in both the foreground and background. These can act as potential occluders.
    - *Identifying features:* Key identifying features may not be fully visible (for instance, a mug may be occluded by distractors such that the logo is only partially visible.)
- **Pose:**
    - *Background:* The object is against different background from the Train scene.
    - *Pose*: Object is in a different pose from the training scene.
    - *Distractors:* No similarly-sized distractor objects nearby, or in the clear foreground.
    - *Identifying features:* Key identifying features may not be fully visible (for instance, a mug may be turned so that its logo is only partially visible.)
- **Both:**
    - *Background:* The object is against different background from the Train scene.
    - *Pose*: Object is in a different pose from the training scene.
    - *Distractors:* Object is surrounded by 2-5 distractor objects of varying sizes, located in both the foreground and background. These can act as potential occluders.
    - *Identifying features:* Key identifying features may not be fully visible (for instance, a mug may be turned or occluded so that its logo is only partially visible.)

For the final (hardest) scene, we attempt to capture images that mimic expected in-the-wild settings for each category. For instance, "Both" scenes for shoes are captured outdoors, with sports equipment as distractors. Similarly mugs are captured on a drying rack, with other wares as distractors.

**Capture.** We capture all images on an iPhone 15 Pro. For each scene, we use the PolyCam app to capture a video. The app automatically extracts approximately 20 frames to be exported. We capture each video in three 120-180 degree views, each at a different vantage point: Level with the object, above the object (camera looking down), below the object (camera looking up). Note that this results in images where the object is not centered, occluded by distractor objects, or out of focus in the background; these are useful as hard positives.

**Splits and annotation.** For each object, we manually inspect the Training scene and extract three training images, taken at level with the target object, and roughly equally spaced throughout the camera trajectory. The rest of the Training scene images are relegated to the test set, and serve as the in-distribution split. Images from the other three scenes serve as the three out-of-distribution (OOD) splits.

**Annotation.** We record a unique identifier for each object, and label every test image with the identifier of the object in that image. These image-level labels are used for classification and retrieval.

For each object, we randomly choose 3 images from each test scene to annotate with segmentation masks and bounding boxes. Thus there are 12 images total per object with dense annotations. To annotate with masks, we first use Grounding-SAM Kirillov et al. (2023a) to generate mask proposals. We then manually inspect every image. For images with incorrect generated masks, we manually annotate using TORAS (Kar et al., 2021).

### A.3 PRIVACY AND CONSENT

**PODS.** Our PODS dataset is published under the MIT License. The majority of objects belong to the authors, and for additional objects, we obtained explicit consent from their owners. We ensured that no personal-identifying attributes are present and that the dataset contains only object images, with no human subjects.

**Dogs.** The DogFaceNet dataset is under the MIT License, allowing modification and redistribution. Our reformulation follows this license with proper attribution.

**DF2.** For DeepFashion2, we comply with the dataset's terms of use by using it solely for non-commercial research purposes without reproducing or redistributing the data. Our reformulation is made public only as a mapping to the original dataset, ensuring that we do not share the data directly.

### A.4 SYNTHETIC DATA GENERATION

#### A.4.1 LLM-GENERATED CAPTIONS

We generated 100 instance-relevant prompts for image generation, which we used to generate images. Here we present 15 examples for each dataset. The full caption set is in our Github repository.

**PODS Dataset**

- Mugs
    1. A `<new1>` mug on a wooden desk
    2. A `<new1>` mug in a cozy living room
    3. A `<new1>` mug on a windowsill
- Bottles
    1. A `<new1>` bottle on a picnic table
    2. A `<new1>` bottle in a backpack pocket
    3. A `<new1>` bottle on a yoga mat
- Screwdrivers
    1. A `<new1>` screwdriver in a toolbox
    2. A `<new1>` screwdriver on a wooden workbench

    3. A `<new1>` screwdriver in a carpenter's tool belt

- Totes (Bags)

    1. A `<new1>` bag in a car trunk

    2. A `<new1>` bag on a park bench

    3. A `<new1>` bag in a shopping cart

    4. A `<new1>` bag on a library shelf

    5. A `<new1>` bag in a gym locker

    6. A `<new1>` bag on a wooden table

- Shoes

    1. A `<new1>` shoe in the rain

    2. A `<new1>` shoe on a sandy beach

    3. A `<new1>` shoe in a gym locker

**DF2 Dataset**

1. A person wearing a `<new1>` shirt at a park
2. A `<new1>` shirt on a mannequin
3. A person wearing a `<new1>` shirt at a party
4. A `<new1>` shirt on a clothesline
5. A person wearing a `<new1>` shirt at a concert
6. A `<new1>` shirt on a chair
7. A person wearing a `<new1>` shirt at a café
8. A `<new1>` shirt on a laundry basket
9. A person wearing a `<new1>` shirt at a stadium
10. A `<new1>` shirt on a hook
11. A person wearing a `<new1>` shirt at a bus stop
12. A `<new1>` shirt on a drying rack
13. A person wearing a `<new1>` shirt at a gym
14. A `<new1>` shirt on a shelf
15. A person wearing a `<new1>` shirt at a picnic

**Dogs Dataset**

1. A `<new1>` dog in the park
2. A `<new1>` dog at the vet
3. A `<new1>` dog in a car
4. A `<new1>` dog at the groomer
5. A `<new1>` dog on a walk
6. A `<new1>` dog in the snow
7. A `<new1>` dog at the lake
8. A `<new1>` dog in the backyard
9. A `<new1>` dog at the `<new1>` dog park
10. A `<new1>` dog in a sweater
11. A `<new1>` dog in a bed
12. A `<new1>` dog at the farm
13. A `<new1>` dog in the woods
14. A `<new1>` dog in a kennel
15. A `<new1>` dog at a barbecue

### A.4.2 MASKED DREAMBOOTH - FILTERING

We apply automatic filtering to the Masked DreamBooth pipeline as an additional data-processing step that we can take when masks are available, to ensure high-quality generated data. We use the masks to extract a bounding box for the object of interest in the training image, and embed it using DreamSim (Fu et al., 2023), a perceptual similarity metric. Similarly, at every test-image prediction, we apply perSAM to generate a test-mask, and embed the masked test image with DreamSim. A cosine similarity score is computed between the train masked embedding and the test masked embedding, and an empirically chosen threshold is used to filter out the data below a certain threshold value. For DF2 and PODS, the threshold was $0.6$ and for Dogs it was $0.55$.

### A.4.3 CUT AND PASTE

To generate cut and paste images, we first require a small subset of training images and their corresponding masks, which we use to extract the foreground object. For the background, we generate 600 unique background scenes following the same set of LLM-prompts used to generate the diverse DreamBooth images. To use them for background generation, we removed the `"<new1>"` specification from every prompt: e.g. `"photo of a <new1> at the beach"` becomes `"photo of a beach"` and addressed proposition inconsistencies afterwards. We then randomly resized the masked foreground image to a scale between 0.3 and 1.3 times the original image size, and pasted it onto the background-generated images at a randomly selected location within the image.

### A.4.4 RUNTIMES

We report the wall-clock runtimes of synthetic data generation methods, using a single NVIDIA A100 GPU, in Table 3. Per-Image generation times are taken as an average over 50 generations. Per-Dataset times indicate the time to generate a 450-image dataset with a batch size of 1. We do not take into account the time to download open-source datasets (e.g., real backgrounds for Cut/Paste), as this is highly context dependent.

| Method | Fine-tuning (min) | Generation per-Image (sec) | Generation per-Dataset (min) | Total (min) |
|---|---|---|---|---|
| DreamBooth (no filtering) | 3.8 | 0.98 | 7.35 | 11.15 |
| DreamBooth (w/ filtering) | 3.8 | 1.83 | 13.7-152.5 | 17.5-156.3 |
| Cut/Paste (real BG) | - | 0.06 | 0.45 | 0.45 |
| Cut/Paste (generated BG) | - | 1.04 | 7.8 | 7.8 |

Table 3: **Runtime for synthetic data generation.** We report the wall-clock runtimes for generating a 450-image synthetic dataset, broken down by stage. The fastest method is Cut/Paste with real backgrounds, which does not require a T2I model.

## B METHODS

### B.1 TRAINING

We use the following hyperparameters to LoRA fine-tune each backbone:

- Learning rate: 0.0003
- Batch size: 16
- LoRA rank: 16
- LoRA alpha: 0.5
- LoRA dropout: 0.3

### B.2 EVALUATION

**Classification:** We take all the available test examples for each instance of interest as our test set. We evaluate each trained personalized embedding space in a one-vs-all binary classification setting

with respect to the rest of our test data, using a standard few-shot learning setup. Given a test image $\tilde{x}$ we use a frozen vision encoder, $f$, to obtain embeddings of $\tilde{x}$. We then compute the maximum cosine similarity between $f(\tilde{x})$ and any real image in $\mathcal{D}_R$, and take this as the prediction confidence. Samples with a confidence above some threshold $t$ are taken as positives; our evaluation metric is thus the Area under the Precision-Recall Curve, which is a threshold-free metric. The performance for a dataset is computed as an average over the PR-AUC for each learned embedding space. We compare the performance between our learned personalized embedding spaces and non-personalized models (i.e. pretrained DINOv2).

**Retrieval:** We take text images as a query set and all available reference training images as a retrieval set. Given a test image $\tilde{x}$ we use a frozen vision encoder, $f$, to obtain embeddings of $\tilde{x}$. We then compute the maximum cosine similarity between $f(\tilde{x})$ and every image in the retrieval set, $\mathcal{D}_R$, and take this as the prediction ranking for the retrieval task. We score the resultant ranking with the standard NDCG metric, as it considers both relevant and position of all retrieved items, unlike F1 and MRR metrics.

**Segmentation:** To be able to perform localization, we use the encoder's patch embeddings. First, we obtain target local features by computing the masked embedding of a training image. We then compute the cosine similarity between the target local features and the patch embeddings of the target test image to generate a local confidence map, where high confidence regions correspond to localization probability of the object. We then apply binarization directly to the local confidence map with Otsu's thresholding method, and upscale it to the image dimensions to yield a segmentation prediction. We score this prediction by taking the aggregate local confidence map values in the predicted mask region. We evaluate our segmentation task using the standard mask AP metric, and also report f1 scores given the fact that our test sets are highly imbalanced (there are significantly more negatives than objects of interest in the test set).

**Detection:** For detection, we apply the same pipeline as segmentation to obtain a local confidence map and a binarized mask. We extract the bounding box prediction from the prediction by drawing a box around the boundaries segmentation mask. We score this prediction by taking the aggregate local confidence map values in the predicted bounding box region. We evaluate our detection task using the standard AP metric, and also report f1 score.

## C ADDITIONAL EXPERIMENTS

### C.1 HYPERPARAMETER AND LOSS FUNCTION ABLATIONS

We conduct ablations of the key training parameters of our method: # anchor-positive pairs, total # generated positives, and choice of loss function. We do so on the validation set of DF2, using the Masked DreamBooth dataset (without filtering); these were chosen arbitrarily, and intended to be representative of trends that we might expect to see across other datasets.

| | # Synthetic Imgs | # Anchor-Pos Pairs | Classification | Retrieval |
|---|---|---|---|---|
| DINOv2 | - | - | 12.0 | 35.9 |
| DINOv2-P | 300 | 300 | 27.2 | 51.2 |
| | 300 | 600 | 35.9 | 58.7 |
| | 300 | 1500 | 38.0 | 61.1 |
| | 300 | 3000 | 38.1 | **62.1** |
| | 300 | 6000 | **38.5** | 61.7 |
| | 300 | 15000 | 37.7 | 61.3 |
| | 300 | 30000 | 36.9 | 60.0 |

Table 4: **Ablation on the number of generated positives.**

**Number of anchor-positive pairs.** Given a fixed number of synthetic and real images, we can potentially sample many combinations of anchors and positives for contrastive learning. We thus sweep over different ratios of generated positives to sampled anchor-positive pairs. We fix the size

of $\mathcal{D}_S$, the pool of generated positives, to 300 (arbitrarily chosen) and sample increasing numbers of anchor-positive pairs from this pool for training. We find that performance on the DF2 validation set plateaus at a 1:10 ratio, and subsequently use this ratio in all of our main experiments. Results are shown in Table 4.

| | # Synthetic Imgs | # Anchor-Pos Pairs | Classification | Retrieval |
|---|---|---|---|---|
| DINOv2 | - | - | 12.0 | 35.9 |
| DINOv2-P | 3 | 30 | 12.8 | 36.0 |
| | 30 | 300 | 28.6 | 35.9 |
| | 150 | 1500 | 36.8 | 60.8 |
| | 300 | 3000 | 38.1 | **62.1** |
| | 450 | 4500 | **38.7** | 61.9 |
| | 600 | 6000 | 38.3 | 60.8 |

Table 5: **Ablation on the number of anchor-positive pairs.**

**Number of generated positives.** We ablate the size of $\mathcal{D}_S$, the pool of generated positives. To isolate the effect of the positive pool size, we fix the ratio between the size of $\mathcal{D}_S$ and the number of anchor-positive pairs that are sampled from $(\mathcal{D}_R, \mathcal{D}_S)$. We fix this ratio to 1:10, as this was found to be best in the previous ablation. We find that performance plateaus at 450 generated positives and thus use 450 as the size of $\mathcal{D}_S$ in all of our main experiments. Results are shown in Table 5.

**Loss function.** We evaluate DINOv2-P trained on the Masked DreamBooth dataset (CFG 5) with three contrastive loss functions, and one non-contrastive loss. For the Cross-Entropy experiment, we add a linear layer with a sigmoid that projects the output feature vector to a single prediction scalar (1 indicating the target object, 0 for any negative). We evaluate on the validation set of DF2 and find that InfoNCE leads to the best results, and that contrastive losses overall perform better than Cross-Entropy. Results are shown in Table 6.

| | Loss Function | Classification | Retrieval |
|---|---|---|---|
| DINOv2 | - | 12.0 | 35.9 |
| DINOv2-P | InfoNCE | **36.5** | **63.3** |
| | InfoNCE (Multi-Positive) | 27.5 | 28.0 |
| | Hinge | 29.4 | 48.4 |
| | Cross-Entropy | 24.9 | 37.0 |

Table 6: **Loss function ablation.**

## C.2 FULL EVALUATIONS ACROSS SYNTHETIC DATASETS

We ablate components of our method that contribute to the diversity of generated datasets, in particular the CFG parameter, and the use of LLM-generated prompts. In Tables 7-8 we show results across all tested synthetic datasets, backbones, and downstream tasks. We highlight the synthetic datasets that lead to the best performance for each backbone (as determined by average performance across the PODS/DF2/Dogs validation sets). For these, we report the minimum/maximum performance across three seeds, with the averages shown in Table 1. Notably, LLM-generated prompts significantly improve performance on global tasks, however have little impact on dense task performance.

| Model | CFG | LLM | Classification | | | Retrieval | | |
|---|---|---|---|---|---|---|---|---|
| | | | PODS | DF2 | Dogs | PODS | DF2 | Dogs |
| DINOv2 | - | - | 28.1 | 14.4 | 83.1 | 69.6 | 36.3 | 89.4 |
| DINOv2-P | 4 | × | 43.1 | 32.7 | 80.2 | 72.9 | 59.3 | 92.3 |
| | 5 | × | 42.9 | 32.6 | 80.4 | 72.7 | 58.8 | 92.2 |
| | 7.5 | × | 41.5 | 32.0 | 81.4 | 71.6 | 57.9 | 92.4 |
| | 4 | ✓ | 47.4 | 35.7 | 80.7 | 79.8 | 63.4 | 94.3 |
| | 5 | ✓ | (48.0, 48.1) | (35.1, 36.1) | (81.2, 81.8) | (79.3, 80.0) | (64.0, 64.4) | (94.3, 94.9) |
| | 7.5 | ✓ | 48.5 | 36.4 | 82.8 | 79.7 | 62.9 | 94.4 |
| CLIP | - | - | 26.7 | 12.7 | 36.4 | 61.4 | 34.7 | 58.0 |
| CLIP-P | 4 | × | 43.4 | 25.0 | 57.6 | 60.3 | 46.9 | 72.4 |
| | 5 | × | 43.0 | 25.0 | 57.9 | 60.0 | 46.1 | 72.5 |
| | 7.5 | × | 40.3 | 24.4 | 58.2 | 59.1 | 45.5 | 72.7 |
| | 4 | ✓ | 46.9 | 26.8 | 64.2 | 72.2 | 51.2 | 80.5 |
| | 5 | ✓ | (47.1, 47.7) | (26.0, 27.3) | (64.4, 66.0) | (71.3, 71.9) | (50.4, 51.8) | (80.5, 81.6) |
| | 7.5 | ✓ | 48.7 | 25.3 | 65.3 | 71.9 | 49.2 | 81.0 |
| MAE | - | - | 8.7 | 5.2 | 11.3 | 34.6 | 25.8 | 33.6 |
| MAE-P | 4 | × | 15.0 | 14.0 | 25.1 | 28.0 | 24.4 | 35.8 |
| | 5 | × | 14.9 | 14.0 | 26.2 | 27.9 | 23.9 | 36.3 |
| | 7.5 | × | 14.1 | 13.5 | 27.0 | 27.4 | 23.6 | 37.0 |
| | 4 | ✓ | (16.6, 17.5) | (12.0, 12.5) | (29.7, 30.1) | (30.6, 30.8) | (23.7, 23.7) | (42.5, 42.9) |
| | 5 | ✓ | 17.1 | 12.3 | 30.4 | 30.4 | 23.0 | 43.1 |
| | 7.5 | ✓ | 16.7 | 11.2 | 32.3 | 29.8 | 21.9 | 43.8 |

Table 7: **Performance of personalized v. pretrained representations on global tasks.** We report results for all generated synthetic datasets, ablating both CFG and the use of LLM-generated prompts. The best dataset for each backbone (selected using validation performance) is highlighted in yellow, with the min/max performance over 4 seeds reported.

| Model | CFG | LLM | Detection (mAP) | | | Detection (F1) | | | Segmentation (mAP) | | | Segmentation (F1) | | |
|---|---|---|---|---|---|---|---|---|---|---|---|---|---|---|
| | | | PODS | DF2 | Dogs | PODS | DF2 | Dogs | PODS | DF2 | Dogs | PODS | DF2 | Dogs |
| DINOv2 | - | - | 11.0 | 5.3 | 12.2 | 11.1 | 6.6 | 13.9 | 12.9 | 4.7 | 12.5 | 14.5 | 5.7 | 15.6 |
| DINOv2-P | 4 | × | 12.7 | 9.0 | 16.9 | 12.0 | 10.7 | 18.8 | 14.3 | 8.0 | 16.5 | 15.1 | 9.4 | 19.6 |
| | 5 | × | 12.9 | 8.8 | 16.0 | 12.1 | 10.7 | 17.9 | 14.4 | 8.0 | 15.8 | 15.0 | 9.4 | 18.9 |
| | 7.5 | × | 13.0 | 8.6 | 16.2 | 12.4 | 10.3 | 18.3 | 14.4 | 7.8 | 16.0 | 15.1 | 9.1 | 19.1 |
| | 4 | ✓ | 12.1 | 9.9 | 17.3 | 10.7 | 11.3 | 19.5 | 13.5 | 9.3 | 17.2 | 13.5 | 10.4 | 20.8 |
| | 5 | ✓ | (12.4, 13.0) | (9.7, 9.9) | (17.0, 17.7) | (11.0, 11.6) | (11.1, 11.4) | (19.6, 20.2) | (13.6, 13.9) | (9.2, 9.5) | (16.9, 17.3) | (13.4, 14.0) | (10.5, 10.7) | (20.4, 20.9) |
| | 7.5 | ✓ | 13.0 | 10.4 | 16.8 | 11.5 | 11.9 | 19.2 | 14.4 | 9.7 | 16.7 | 14.2 | 10.9 | 20.1 |
| CLIP | - | - | 0.1 | 2.9 | 6.1 | 0.1 | 3.5 | 6.3 | 0.3 | 2.9 | 7.3 | 0.2 | 3.3 | 8.0 |
| CLIP-P | 4 | × | 0.5 | 4.7 | 7.8 | 0.3 | 5.0 | 7.8 | 1.3 | 4.7 | 9.0 | 0.7 | 5.0 | 9.7 |
| | 5 | × | 0.4 | 4.3 | 7.7 | 0.2 | 4.7 | 7.7 | 1.3 | 4.4 | 8.9 | 0.6 | 4.7 | 9.6 |
| | 7.5 | × | 0.8 | 4.3 | 8.1 | 0.4 | 4.7 | 8.0 | 1.4 | 4.4 | 9.3 | 0.7 | 4.8 | 10.0 |
| | 4 | ✓ | 0.9 | 4.5 | 9.0 | 0.5 | 4.8 | 9.3 | 1.7 | 4.4 | 10.2 | 0.9 | 4.7 | 11.0 |
| | 5 | ✓ | (0.7, 0.8) | (4.7, 4.9) | (9.1, 10.0) | (0.4, 0.4) | (5.1, 5.1) | (9.3, 9.8) | (1.5, 1.7) | (4.8, 5.0) | (10.5, 11.3) | (0.8, 0.9) | (5.1, 5.3) | (11.5, 11.9) |
| | 7.5 | ✓ | 0.8 | 4.7 | 9.8 | 0.4 | 5.1 | 9.8 | 1.6 | 4.8 | 10.8 | 0.9 | 5.1 | 11.5 |
| MAE | - | - | 0.2 | 1.4 | 1.4 | 0.2 | 1.9 | 2.1 | 0.4 | 1.2 | 0.7 | 0.3 | 1.6 | 1.0 |
| MAE-P | 4 | × | 0.4 | 1.5 | 1.2 | 0.4 | 2.1 | 1.9 | 0.5 | 1.2 | 0.5 | 0.5 | 1.5 | 0.8 |
| | 5 | × | 0.4 | 1.4 | 1.2 | 0.4 | 2.0 | 1.9 | 0.6 | 1.1 | 0.6 | 0.6 | 1.5 | 0.8 |
| | 7.5 | × | 0.3 | 1.5 | 1.2 | 0.3 | 2.1 | 1.9 | 0.4 | 1.2 | 0.6 | 0.5 | 1.6 | 0.9 |
| | 4 | ✓ | (0.4, 0.5) | (1.5, 1.5) | (1.5, 1.6) | (0.3, 0.4) | (2.0, 2.1) | (2.0, 2.2) | (0.5, 0.6) | (1.2, 1.3) | (0.7, 0.9) | (0.5, 0.5) | (1.5, 1.6) | (0.9, 1.1) |
| | 5 | ✓ | 0.4 | 1.6 | 1.6 | 0.4 | 2.2 | 2.1 | 0.5 | 1.3 | 0.7 | 0.5 | 1.7 | 0.9 |
| | 7.5 | ✓ | 0.5 | 1.7 | 1.6 | 0.5 | 2.3 | 2.1 | 0.6 | 1.2 | 0.8 | 0.6 | 1.6 | 1.0 |

Table 8: **Performance of personalized v. pretrained representations on dense tasks.** We report results for all generated synthetic datasets, ablating both CFG and the use of LLM-generated prompts. The best dataset for each backbone (selected using validation performance) is highlighted in yellow, with the min/max performance over 4 seeds reported.

### C.3 EVALUATION ON PERSAM

Here, we show that our personalized representations can be plugged into state-of-the-art pipelines for personal tasks. We experiment with PerSAM Zhang et al. (2023). In the PerSAM method, an image encoder is used to extract patch features of one or more training images, which are compared to the patch features of a test image to extract a confidence map. The confidence map is then used to generate keypoint proposals to prompt SAM, and guide the attention map of the SAM decoder (refer to Zhang et al. (2023) for details). We evaluate PerSAM with DINOv2, and DINOv2-P as the image encoder, across DF2, Dogs, and PODS. Our results are shown in Table 9; we find that DINOv2-P improves personalized segmentation performance over DINOv2 for all datasets. We use DINOv2-P trained with the best-performing dataset for the DINOv2 backbone (see Tables 7-8).

|  |  | Segmentation (mAP) | Segmentation (F1) |
|---|---|:---:|:---:|
| **PODS** | DINOv2 | 17.1 | 20.8 |
|  | DINOv2-P | 22.3 ↑ | 24.7 ↑ |
| **DF2** | DINOv2 | 4.8 | 5.8 |
|  | DINOv2-P | 10.0 ↑ | 10.9 ↑ |
| **Dogs** | DINOv2 | 16.6 | 20.9 |
|  | DINOv2-P | 21.4 ↑ | 26.2 ↑ |

Table 9: **Application to PerSAM**

### C.4 DIVERSITY AND FIDELITY ANALYSIS

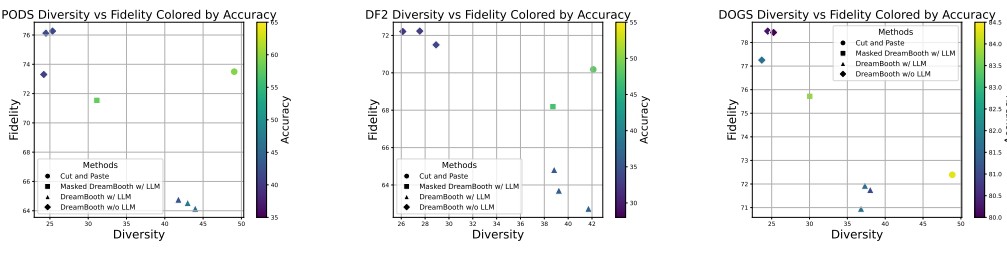

Figure 12: **Diversity and fidelity plot colored by accuracy** across PODS, DF2 and Dogs synthetic datasets. Note that the fidelity metric may be influenced by the background features appearing in the cropped image, resulting in a reduced fidelity score for some samples.

To better understand the characteristics of our generated datasets, we perform a diversity and fidelity analysis in Figure 12. We measure fidelity by computing the DreamSim cosine similarity between synthetic images and the mean embedding of the reference real images, both cropped around the object of interest. To measure diversity, we compute the pairwise similarity between DreamSim features of the synthetic images in each dataset.

Notably, optimal accuracy is achieved when both fidelity and diversity are sufficiently balanced—too much fidelity at the expense of diversity, or too much diversity with low fidelity, both lead to degraded performance. Maintaining appropriate levels of both results in the best performance, and this can be achieved in our work through Masked DreamBooth or Cut and Paste generative methods.

## D QUALITATIVE RESULTS

### D.1 DENSE PREDICTION VISUALIZATIONS

In Figures 13-15 we visualize dense predictions with personalized patch features for randomly sampled images/classes.

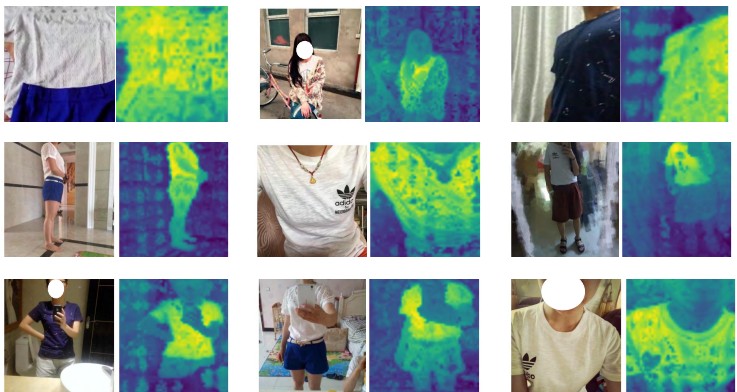

Figure 13: **DF2 Dense Predictions** Images and classes are randomly sampled

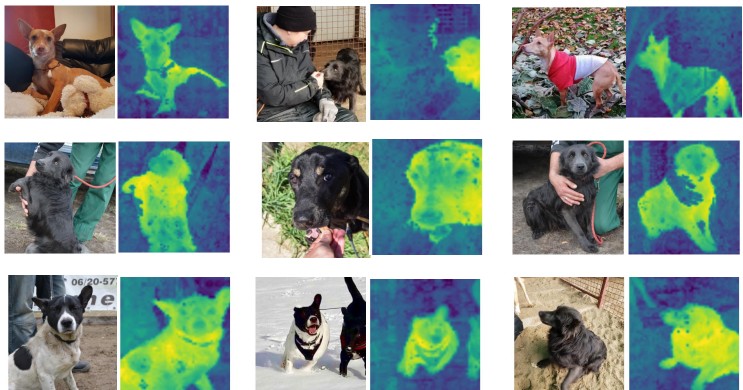

Figure 14: **Dogs Dense Predictions** Images and classes are randomly sampled

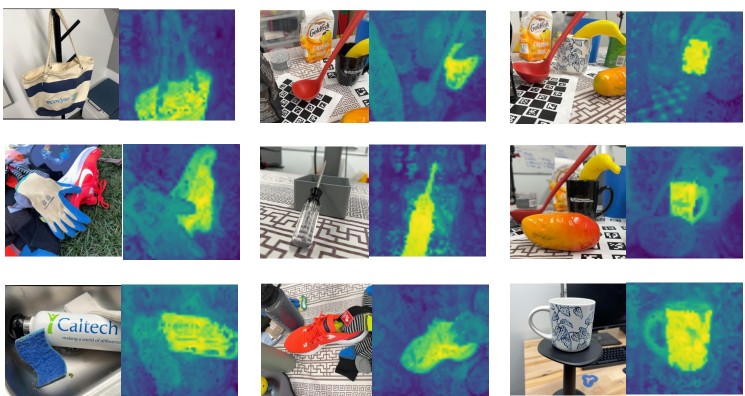

Figure 15: **PODS Dense Predictions** Images and classes are randomly sampled

## D.2 CHALLENGING EXAMPLES

We visualize examples of hard negatives and hard positives from the Dogs dataset to better understand the capabilities of personalized representations. For a given query image $x$ of a target object, we identify hard positives as the $k$ positives with the lowest DINOv2 cosine similarity to $x$ (often the dog in a very different setting or position), and hard negatives as the $k$ negatives with the highest

DINOv2 cosine similarity to $x$ (often different dogs of the same breed). We denote DINOv2-P as successful on a hard negative (positive) if it has a lower (higher) cosine similarity to $x$ than DINOv2.

We show randomly selected examples of hard positives/negatives in the Dogs dataset in Figures 16 and 17 respectively. DINOv2-P is typically successful on hard positives; the cosine similarity between positive pairs nearly always increases, even in cases with significant differences (lighting, pose, etc) from the query image. However, we also identify several cases in which the cosine similarity between query images and hard negatives also increases. This is a failure case that may be induced by noisy positives in the synthetic training dataset, leading the personalized representation to associate the target object with spurious features.

Figure 16: **Dogs Hard Positives.** Given images of a target dog (leftmost of each row) we identify the positive test images with the lowest DINOv2 similarity to the query. DINOv2-P cosine similarity typically increases, even for cases with significant differences in setting, camera angle, occlusion, etc.

## D.3 DREAMBOOTH GENERATED DATA

We present qualitative examples for the following datasets:

- **PODS**: DreamBooth without LLM (Figure 18), PODS DreamBooth with LLM (Figure 19), PODS Masked DreamBooth with LLM + Filtering (Figure 19), PODS negatives (Figure 21).
- **DF2**: DreamBooth without LLM 22, DF2 DreamBooth with LLM 23, DF2 Masked DreamBooth with LLM + Filtering 24, DF2 negatives 25
- **Dogs:** DreamBooth without LLM (Figure 26), DreamBooth with LLM (Figure 27), Masked DreamBooth with LLM + Filtering (Figure 28) and synthetic negatives (Figure 29)

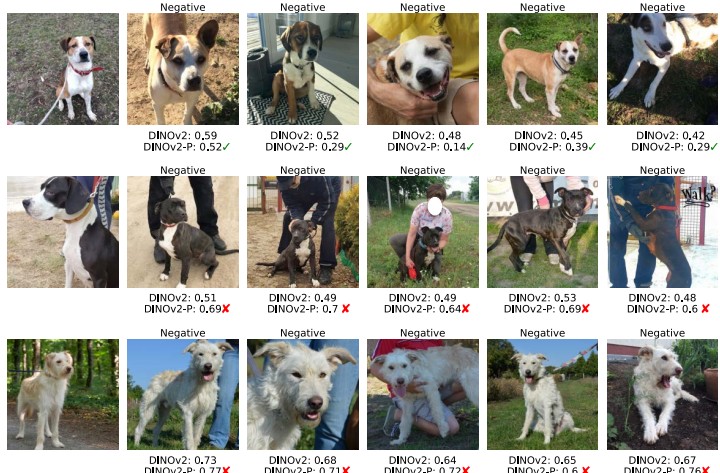

Figure 17: **Dogs Hard Negatives.** Given images of a target dog (leftmost of each row) we identify the negative test images with the highest DINOv2 similarity to the query. In some cases DINOv2-P cosine similarity decreases (top row) however we also identify failure cases where cosine similarity increases (second/third rows).

**PODS Generated Images**

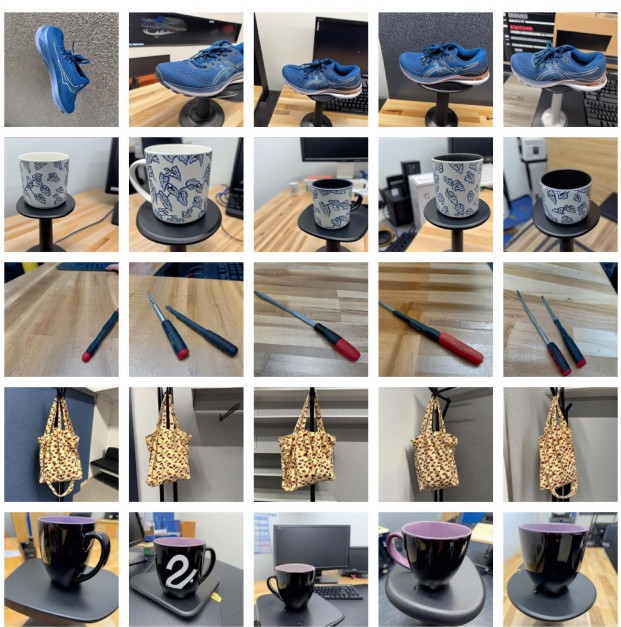

Figure 18: **PODS Generated Images - DreamBooth without LLM prompting.** Images and classes are randomly sampled

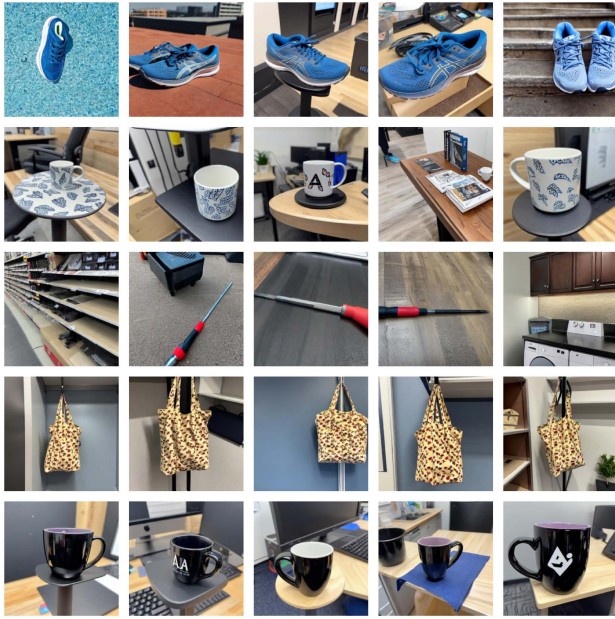

Figure 19: **PODS Generated Images - DreamBooth with LLM prompting.** Images and classes are randomly sampled

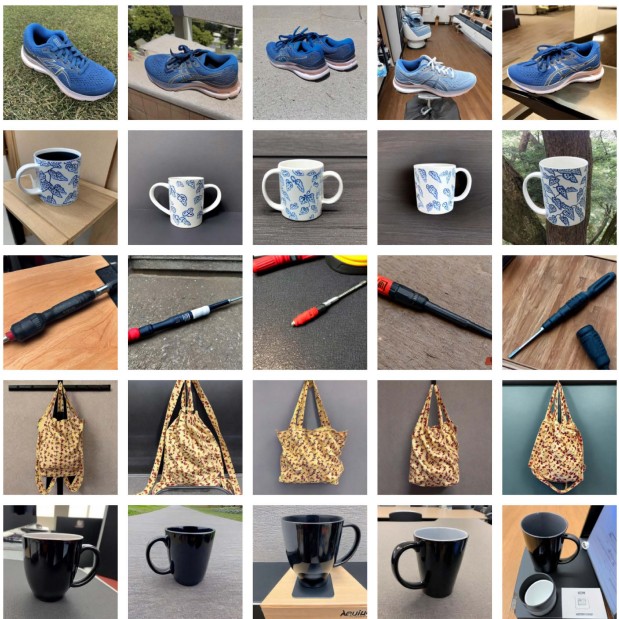

Figure 20: **PODS Generated Images - DreamBooth with LLM, Masking and Filtering.** Images and classes are randomly sampled

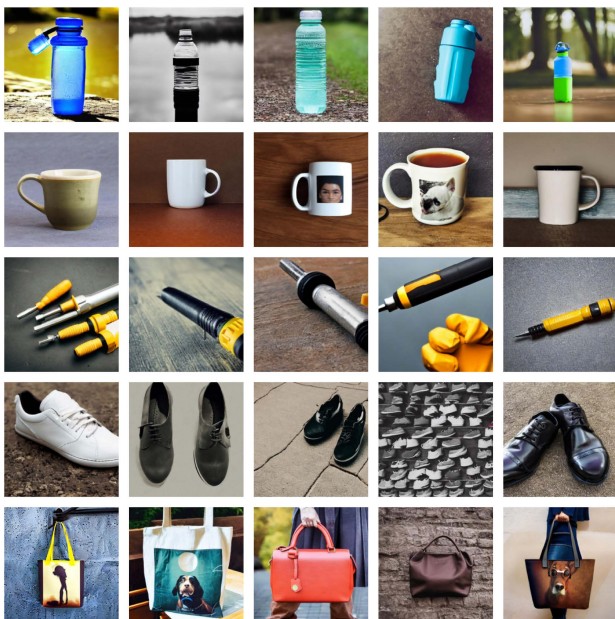

Figure 21: **PODS Generated Negatives.** Images and classes are randomly sampled. Each row are sampled negatives for each object category.

**DeepFashion2 Generated Images**

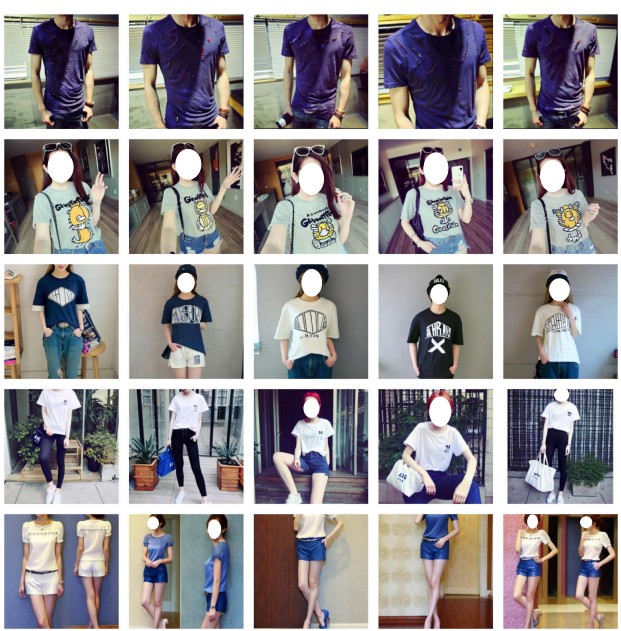

Figure 22: **DF2 Generated Images - DreamBooth without LLM** Images and classes are randomly sampled

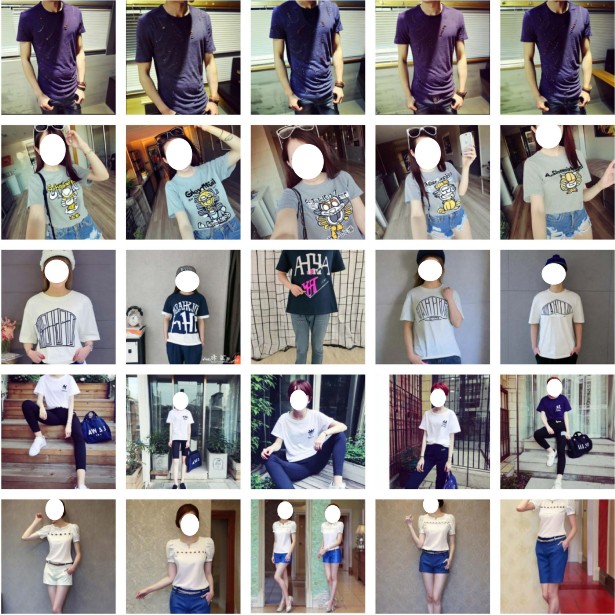

Figure 23: **DF2 Generated Images - DreamBooth with LLM** Images and classes are randomly sampled

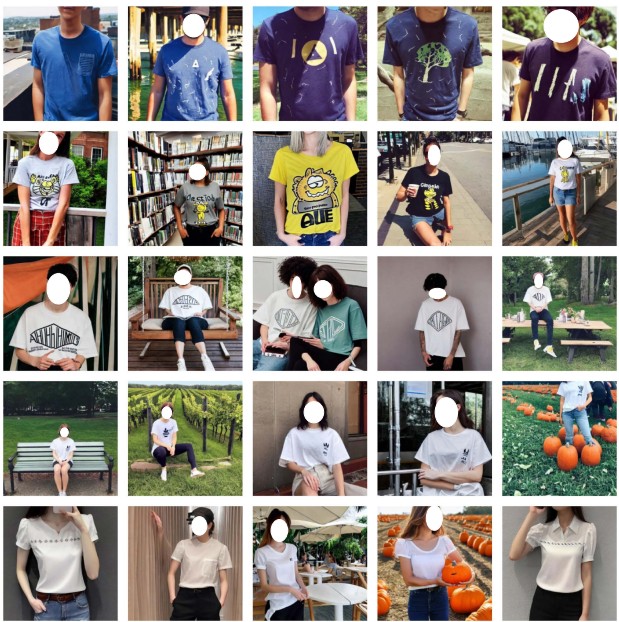

Figure 24: **DF2 Generated Images - DreamBooth with LLM, Masking and Filtering.** Images and classes are randomly sampled

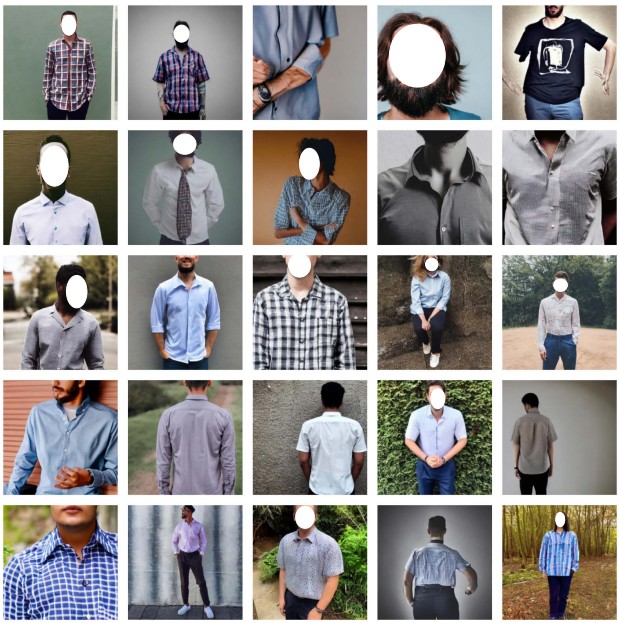

Figure 25: **DF2 Generated Negatives.** Images and classes are randomly sampled

**Dogs Generated Images**

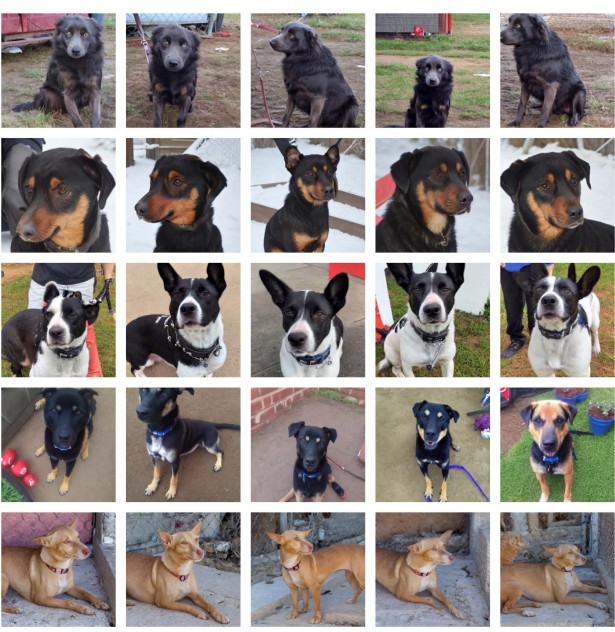

Figure 26: **Dogs Generated Images - DreamBooth without LLM prompting.** Images and classes are randomly sampled

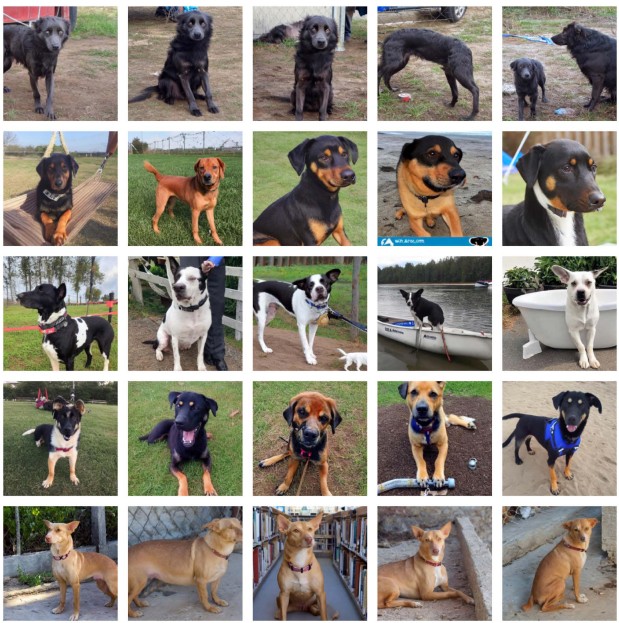

Figure 27: **Dogs Generated Images - DreamBooth with LLM prompting.** Images and classes are randomly sampled

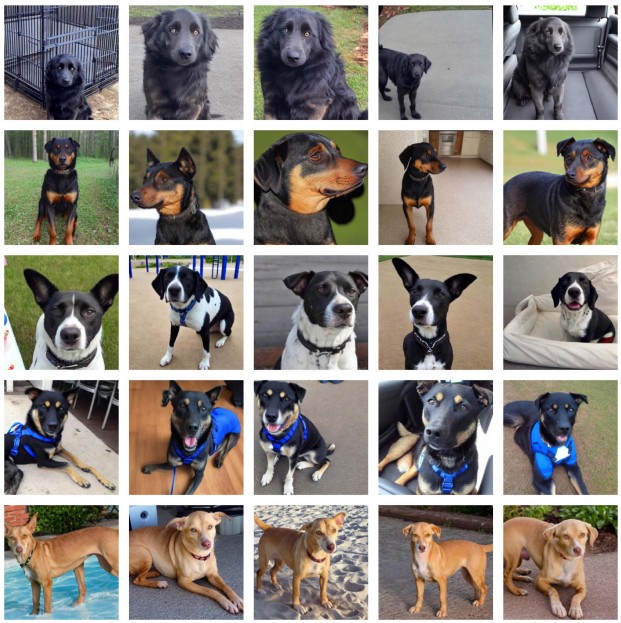

Figure 28: **Dogs Generated Images - DreamBooth with LLM, Masking and Filtering.** Images and classes are randomly sampled

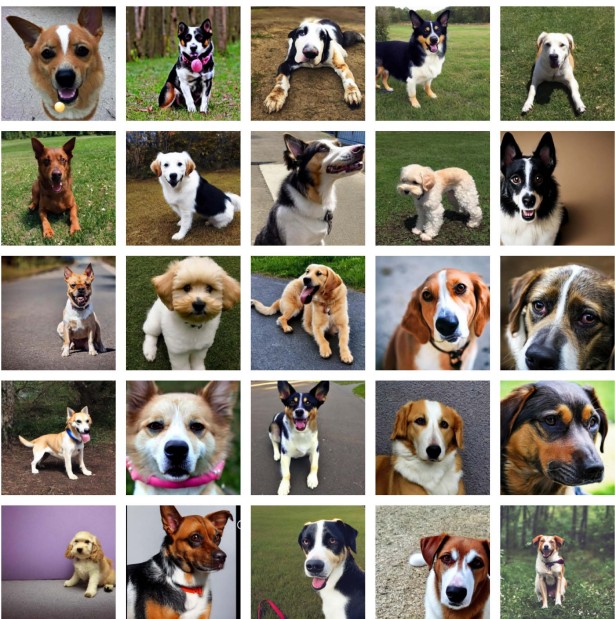

Figure 29: **Dogs Generated Negatives.** Images and classes are randomly sampled

