# OpenReview forum: "Personalized Representation from Personalized Generation"
_ICLR.cc/2025/Conference — ICLR 2025 Poster_

### Official Review · Reviewer_eQii · 2024-10-28

**Soundness:** 2
**Presentation:** 2
**Contribution:** 3
**Rating:** 6
**Confidence:** 4

**Summary:**

This paper addresses the challenge of using modern vision models for personalized vision tasks, which are often fine-grained and limited in data availability. The authors propose a method that combines insights from synthetic data in general representation learning and advances in T2I (text-to-image) diffusion models, leveraging personalized synthetic data to learn representations specific to an object of interest. To support this goal, they introduce an evaluation suite consisting of reformulated datasets and a novel dataset designed for personalized learning. The proposed contrastive learning approach uses image generators to enhance representation learning for diverse downstream tasks, such as recognition and segmentation.

**Strengths:**

- Clarity and Structure: The paper is well-organized and clearly written, making it accessible and easy to follow, even for readers who may be less familiar with the technical aspects of personalized representation learning in vision models.
- Visualization Quality: The visualizations of generated images are well-designed and effectively demonstrate the model’s representation capabilities, enhancing the clarity and impact of the experimental results.
- Meaningful Personalized Representation: The concept of learning personalized representations for specific objects is a valuable contribution, offering potential applications across various fine-grained and data-scarce tasks where general-purpose representations may fall short.

**Weaknesses:**

- Effectiveness of contrastive learning: It is unclear if the observed performance gains stem from the proposed method or from the inherent knowledge in Stable Diffusion 1.5. In the appendix, “Table 4: Ablation on the Number of Anchor-Positive Pairs” indicates poor performance when the “# Synthetic Imgs” and “# Anchor-Pos Pairs” are small. Could the authors provide evidence that the improvement in representation learning is not primarily due to the knowledge embedded in Stable Diffusion? A comparison between the proposed contrastive approach and straightforward fine-tuning with the same number of synthetic positives (maybe I2T+LLM+Stable Diffusion) would help clarify this point.
- Contrastive Learning Setup and Objectives: The purpose and setting of the contrastive learning approach remain ambiguous. If the objective is solely to improve personalized representation at inference, it would be useful to include more baselines to demonstrate the effectiveness of your contrastive learning, such as using simple LORA/finetuning on positives obtained from I2T+LLM+Stable Diffusion. Conversely, if the goal includes a defensive aspect, ensuring that performance declines for objects other than the target, results showing the performance on unrelated objects would provide valuable context.
- Typos and Formatting:
  - Line 53: "e.g. a model" should be "e.g., a model".
  - Line 59: "e.g. recognizing" should be "e.g, recognizing".
  - Line 78: "i.e. no" should be "i.e., no".
  - Line 153: "e.g. segmentation" should be "e.g., segmentation".
  - Line 162: "e.g. one" should be "e.g., one".
  - Line 348: ”query” and ”retrieval” should be formatted as ``query``” and “``retrieval``.
  - Figure 2: Missing period. "Pipeline Our" should be "Pipeline. Our".
  - Figure 5: "following 4. " should be "following Figure 4. ".

**Questions:**

- Distillation vs. Personalized Fine-Tuning: Could the authors clarify whether the observed performance gains primarily result from their proposed contrastive method or from pre-existing knowledge in Stable Diffusion 1.5? Would a comparison with a straightforward fine-tuning approach on a large set of synthetic positives help isolate the benefits of the proposed method?
- Contrastive Learning Setup and Objectives: Regarding the contrastive learning approach, could the authors elaborate on its intended objectives? If the primary aim is to enhance personalized representation capabilities at inference, would adding a baseline using simple LORA or finetuning with I2T+LLM+Stable Diffusion positives be beneficial? Alternatively, if there is a defensive objective to reduce performance for non-target objects, would the authors consider providing results on unrelated objects?
- Clarification of Typos and Formatting: There are minor typos and formatting inconsistencies throughout the text (e.g., “e.g.” should be formatted as “e.g.,”). Would the authors consider a thorough review to address these to improve readability?

**Details Of Ethics Concerns:**

The paper introduces an evaluation suite tailored to the challenge, which includes reformulated versions of two existing datasets as well as a novel dataset constructed specifically for this study. Further clarification on any privacy or consent measures in place for the datasets would be beneficial for a full understanding of ethical compliance.

---

> ### Author Response · Authors · 2024-11-21
>
> Thank you for your insightful, helpful feedback. We have addressed your comments below; please let us know if you have any further questions.
>
> > ***Distillation v. Personalized finetuning***
>
> Thank you for raising this point – it is important to isolate the source of performance gains (to clarify, we do not introduce a new loss function, and rely on the existing infoNCE function).
>
> To separate the effects of the contrastive loss from the synthetic data, **we add cross-entropy, a simple non-contrastive loss, to Table 6**. We initialize a linear layer with a sigmoid function that projects a backbone's output (CLS token concatenated with avg-pooled patch tokens) to a binary prediction (1 indicates the presence of the target object, 0 otherwise). We then supervise with cross-entropy and fine-tune the backbone with LoRA while fully updating the linear layer. On the DF2 validation set, cross-entropy leads to worse results than all contrastive losses. This indicates that a contrastive loss is important for performance gains.
>
> As you note, however, the generative prior in Stable Diffusion is important for generating positives in novel poses and backgrounds. For instance, in Sec 5.3 we show that pose diversity in generations results in downstream robustness to pose variation. This could indeed be considered a form of distillation, and is an advantage of using a T2I generator.
>
> > ***Contrastive learning setup/objectives***
>
> Our objective is to improve a representation's performance for personalized tasks. We chose a contrastive framework for the following reasons:
> * It does not require extra labels. We formulate positive pairs as $(X, g(X))$, where $g$ is a generator, and obtain negatives from the random distribution of images. We then train to enforce representational invariance to the changes (pose, background, etc) introduced by $g$.
> * It does not require us to learn additional decoders or heads, which is useful in our low-data regime.
> * Contrastive losses are robust to noise [1]. DreamBooth often fails to preserve fine-grained details, so it is useful to choose a loss that can still extract signal even if the positive pair does not exactly match
>
> As detailed in the previous section, following your suggestion we compare to straightforward cross-entropy supervision (non-contrastive). **Our results suggest that contrastive losses are indeed important for our objectives**.
>
> > ***Typos/formatting***
>
> Thank you for pointing out these typos; **we have corrected them in our revision**.
>
> > ***Ethics review***
>
> Below we give details on ethics/privacy guidelines for the datasets. Note that for DF2 and Dogs, our reformulation solely entails resplitting the existing published datasets and annotating with masks. We also provide the licenses that both DF2 and Dogs were released under in Sec. A.1.
> * **PODS:** The majority of PODS objects belong to the authors. For all additional objects, we obtained explicit consent from the owners. We chose objects with no personal-identifying attributes (such as the owner's name). We also only include images of objects, with no human subjects.
> * **DF2:** The DeepFashion2 images are taken from the original DeepFashion dataset [2], and online shopping websites [3]. These two benchmarks are widely used/cited in vision literature. To access the dataset we signed an agreement that it only be used for non-commercial research purposes.
> * **Dogs:** The DogFaceNet are images captured by the authors, and downloaded from publicly-available dog photos on adoption websites [4]. The authors do not, to our knowledge, provide further details regarding privacy/consent measures.
>
> [1] Yihao Xue et al. "Investigating why contrastive learning benefits robustness against label noise". ICML, 2022.
>
> [2] Yuying Ge et al. "DeepFashion2: A Versatile Benchmark for Detection, Pose Estimation, Segmentation and Re-Identification of Clothing Images." CVPR, 2019.
>
> [3] Ziwei Liu et al. "DeepFashion: Powering Robust Clothes Recognition and Retrieval with Rich Annotations." CVPR, 2016.
>
> [4] Mougeot, G. et al. "A Deep Learning Approach for Dog Face Verification and Recognition." PRICAI 2019: Trends in Artificial Intelligence, 2019.

---

> > ### Author Response · Authors · 2024-11-25
> >
> > Dear Reviewer eQii,
> >
> > Thank you again for your detailed review! We have added results with straightforward fine-tuning, clarification on distillation v. fine-tuning, and addressed typos in our responses and revision, which we hope sufficiently address your questions/concerns. In the remaining short time, we would be very happy to alleviate any remaining concerns you still have about our paper.
> >
> > Thank you once again for dedicating your time and effort to reviewing our work and providing us with insightful suggestions!

---

> > > ### Author Response · Authors · 2024-11-30
> > > **A Kind Reminder**
> > >
> > > Dear Reviewer eQii,
> > >
> > > Thank you again for taking your time to review our paper! Your feedback was very constructive and we believe that we have addressed all of the concerns/questions in your review, and updated our paper accordingly:
> > > - **Effectiveness of contrastive learning / distillation v. personalized fine-tuning concern:** addressed by adding a non-contrastive loss to our learning objective ablation (**Table 6**), and we also clarify that generative prior in Stable Diffusion, in addition to the contrastive pipeline, plays a role in effective personalized representation learning
> > > - **Contrastive learning setup:** Provided additional justifications around the contrastive objective - does not require extra labels, no additional decoders or heads, and robust to noise (see our initial rebuttal above for details)
> > > - **Corrected all typos/formatting** reviewers have brought to our attention
> > > - Provide detailed **ethics/privacy guidelines** for the datasets
> > >
> > > For detailed discussion, please refer to our response above and the Global Response.
> > >
> > > As we approach the end of the discussion period, please let us know if you have further questions, or if there is anything else we can address for you to consider raising our score. Thank you again for your time and effort in reviewing our paper!

---

### Official Review · Reviewer_2ShX · 2024-11-03

**Soundness:** 3
**Presentation:** 2
**Contribution:** 2
**Rating:** 5
**Confidence:** 4

**Summary:**

The paper explores the use of synthetic data to train personalized visual representations, addressing data scarcity challenges in fine-grained, instance-specific vision tasks. The authors propose a three-stage pipeline involving synthetic image generation, contrastive learning, and fine-tuning of a general-purpose pretrained model using only three real examples of an object. They introduce an evaluation suite, including two reformulated datasets (DeepFashion2 and DogFaceNet) and a novel PODS dataset, to evaluate the effectiveness of their approach across tasks such as classification, retrieval, detection, and segmentation. Results suggest that the proposed method outperforms pretrained models on these tasks and provides an analysis of the impact of generative models (e.g., DreamBooth) on representation quality. However, this work lacks certain methodological and experimental rigor, as discussed in the critique below.

**Strengths:**

The paper presents an interesting approach to personalized visual representation learning using synthetic data, offering a novel perspective on addressing data scarcity in fine-grained instance-specific tasks.
The study showcases the potential of synthetic data in personalized representation learning, especially for data-limited tasks, indicating that personalized generative models could contribute to a broader field of personalized AI applications.
The PODS dataset is a valuable addition to the field, providing a benchmark for evaluating the performance of personalized visual models across multiple tasks. Although limited, it may be useful in future research for controlled comparisons in low-data personalized settings.

**Weaknesses:**

High Computational Cost: The approach’s dependency on DreamBooth and other costly generative models for fine-tuning makes it impractical for widespread use. By not considering more computationally feasible alternatives, such as GAN-based models, the method lacks the flexibility needed for real-world adoption. Lack of Baseline Comparisons and Systematic Experimentation: The paper does not provide enough rigorous comparisons against simpler, cheaper baselines that could yield similar or competitive results, such as training with real, small datasets or using augmentation strategies on a limited set of real images. This gap leaves the reader questioning the unique benefit of using complex synthetic data pipelines, particularly in relation to common data-efficient techniques in computer vision.
Limited Task Scope and Lack of Real-World Validation: Although the paper claims improvements across multiple tasks, the experimental setup is relatively narrow, focusing on a limited range of controlled settings. Given that the proposed approach aims for personalization, testing its robustness in diverse and complex real-world scenarios is essential. Additionally, the observed performance gains in detection and segmentation tasks are relatively small, raising concerns about the method's robustness. Ethical and Privacy Considerations: While the authors briefly address these considerations, they do not offer practical guidelines or safeguards to mitigate potential misuse. Expanding on ethical implications and providing specific measures for responsible deployment would improve the paper’s comprehensiveness. Inadequate Analysis of Data Diversity: Although the authors highlight the role of data diversity in improving representation quality, they fail to conduct a thorough, quantitative analysis of the impact of diversity on representation learning. The paper lacks a clear quantification of diversity’s effect on model performance, and only discusses a few configuration parameters, which does not sufficiently explore the potential of synthetic data augmentation.

**Questions:**

Given the reliance on DreamBooth, which is computationally intensive, have the authors considered alternative  generative models  to achieve similar outcomes?
How does the computational cost compare to simpler data augmentation methods, and what trade-offs are involved in model performance?
Has the study examined the potential biases and limitations introduced by synthetic data in the personalized learning setting, especially in comparison to real data? Could the authors provide any quantitative result to show how the synthetic data representations compare with those learned from real data, particularly in terms of generalization?
The proposed approach has clear applications in sensitive areas, such as surveillance or individual tracking, raising potential privacy concerns. How do the authors propose to mitigate these risks,?

---

> ### Author Response · Authors · 2024-11-21
>
> Thank you for your detailed feedback. We have addressed your comments below; please let us know if you have any further questions.
>
> >***High computational cost & cheaper alternatives***
>
> You raise a good point on cost -- in **Sec. 1 of our Global Response** we discuss new evaluations of cheap alternatives to diffusion models.
>
> We now cover:
> * A **limited** set of real images
> * A **larger** set of real images
> * Synthetic positives **without T2I models**
> * Synthetic positives **with T2I models**
> * Incorporating **internet-available images**
>
> We investigated using GANs, but did not find suitable customized generation methods. However, we hope our new comparisons (see Global Response) provide examples of achieving good performance with cheap alternatives to T2I models.
>
> **In Sec. 2-3 of our Global Response we discuss method runtimes and tradeoffs between using T2I models and cheaper approaches.**
>
> >***Baseline comparisons***
>
> Thank you for your suggestions. We address this above and in **Sec. 1 of our Global Response**; please let us know if further follow-up is needed.
>
> >***Limited task scope / real-world validation***
>
> We completely agree that testing robustness in diverse/complex real-world scenarios is critical. We highlight some ways that our datasets test this:
>
> * **Subjects:** They cover a wide range of subjects: fashion (DF2), animals (Dogs), and household items (mugs, screwdrivers, bags, bottles, shoes).
> * **Robustness/generalization:**
>   * **DF2** is designed to include distribution shift. Training images are store/gallery images, and test images are in-the-wild consumer photos with a variety of settings, poses, distractors, and quality/occlusion.
>   * **PODS:** In Sec. A.2.2 we explain the distribution shifts in PODS. Training images are taken in controlled, simple settings featuring the object one pose/location. Test sets evaluate combinations of variations in pose, background, # objects. We also vary camera angle and occlusion/visibility.
> * **Complex/real-world scenarios:**
>   * **DF2/Dogs:** Images from DF2 and Dogs are collected in the wild, in various environments. These naturally collected images reflect complex, uncurated settings, with no specific control over backgrounds, lighting, or other factors.
>   * **PODS:** The “Both” test subset includes complex/real-world scenarios (see end of A.2.2). For example, mugs are captured on full dish racks, and screwdrivers captured in toolboxes.
>
> **We have added examples from DF2/Dogs (Fig. 10-11) highlighting these aspects.**
>
> >***Biases/limitations in synthetic data***
>
> You are correct that synthetic data introduces bias/limitations. For example, Stable Diffusion struggles with certain objects, such as screwdrivers, leading to worse performance on those categories. It also exhibits a well-known bias towards single-object images, potentially causing lower performance on distractor variations in Fig. 9. In **Section 5.3**, we analyze limitations introduced by DreamBooth and Cut/Paste (which uses real images of the object).
>
> >***Diversity analysis***
>
> Thank you for your feedback – following your suggestion, we added a **quantitative analysis of fidelity / diversity across synthetic datasets (Fig. 12)**, where we plot fidelity v. diversity, colored by performance, for various DreamBooth datasets and Cut/Paste. Non-filtered DreamBooth datasets lie at the extremes (high-diversity, low-fidelity or vice versa); the filtered dataset achieves a better balance of the two, and thus higher performance.
>
> >***How do synthetic-data representations compare to real-data representations in terms of generalization?***
>
> **We updated Fig. 9** to explicitly compare synthetic-learned representations (Masked DreamBooth, Combined DreamBooth + Cut/Paste, Cut/Paste) to real-data-learned representations (Real-Image) on the PODS distribution shifts. Synthetic-augmented datasets generalize significantly better across OOD test sets.
>
> >***Ethical/Privacy concerns***
>
> Thank you for raising this important concern. We acknowledge the potential privacy implications of instance recognition tasks, particularly in sensitive areas (i.e., surveillance). To address this, we deliberately focused on non-human ID tasks and did not collect any human data in the PODS dataset. We recommend that practitioners prioritize ethical deployment by using anonymized data and adhering to standard privacy-preserving guidelines. We also believe that there are significant positive applications when deployed responsibly **(see Global Response Section 2)**.

---

> > ### Author Response · Authors · 2024-11-25
> >
> > Dear Reviewer 2ShX,
> >
> > Thank you again for your detailed review! We have added results/discussion on computationally cheap comparisons, additional baselines, biases/limitations of synthetic data v. real data, generalization capabilities, real-world evaluation, diversity analysis, and ethical/privacy risks. We hope these sufficiently address your questions/concerns. In the remaining short time, we would be very happy to alleviate any remaining concerns you still have about our paper.
> >
> > Thank you once again for dedicating your time and effort to reviewing our work and providing us with insightful suggestions!

---

> > > ### Author Response · Authors · 2024-11-30
> > > **A Kind Reminder**
> > >
> > > Dear Reviewer 2ShX,
> > >
> > > Thank you again for taking your time to review our paper! Your feedback was very constructive and we believe that we have addressed all of the concerns/questions in your review, and updated our paper accordingly:
> > > - **High Computational Cost:** Added runtime information for all methods in **Table 3** and found that there exists a speed v. performance tradeoff. However performance gains are also achievable without T2I models (efficiently). We add this discussion to our **results section in 5.2**, and clarify our contribution wrt cost by updating the **introduction** of our paper
> > > - **Lack of Baseline Comparisons:** We significantly extend our comparisons/baselines to study the tradeoffs (**Table 2** and **Sec. 5.2** of the paper) between generated images and cheaper alternatives that leverage additional data/annotations. We add two new cases: 1) internet-available data and 2) scaling real positives, which allowed us to explore 5 additional baselines across different generative methods in our revision.
> > > - **Limited task scope / real-world validation / generalization:** We provide further descriptions (**Section 4.1**) and qualitative images of the three datasets (**Section A.1**) used in our work to show that they sufficiently benchmark complex real-world scenarios and evaluate generalization. We also update **Fig. 9** to show that synthetic-augmented datasets generalize significantly better than real-image-only across OOD test sets.
> > > - **Biases/limitations in synthetic data:** We acknowledge that synthetic data can introduce bias/limitations. We highlight that in **Section 5.3** we analyze limitations in our downstream representations introduced by synthetic data limitations.
> > > - **Diversity analysis:** We add this quantitative analysis to **Fig. 12** which shows that datasets that achieve a good balance of the two lead to optimal performance
> > > - **Ethical Privacy Concerns:** We acknowledge this concern and share steps we have taken to mitigate unethical practice in our initial comment above.
> > >
> > > For detailed discussion, please refer to our response above and the Global Response.
> > >
> > > As we approach the end of the discussion period, please let us know if you have further questions, or if there is anything else we can address for you to consider raising our score. Thank you again for your time and effort in reviewing our paper!

---

### Official Review · Reviewer_FvPe · 2024-11-03

**Soundness:** 3
**Presentation:** 3
**Contribution:** 3
**Rating:** 6
**Confidence:** 4

**Summary:**

This paper addresses the challenge of learning personalized representations. The authors utilize a generative model to augment the dataset for the target personalized object, followed by fine-tuning a pretrained model using contrastive learning. This approach aims to develop a model capable of handling downstream tasks related to the specific object. Several downstream tasks were evaluated, and the proposed method demonstrated an average improvement across these tasks.

**Strengths:**

1. The paper explores a novel and intriguing problem: learning personalized representations, which could be valuable for downstream tasks related to target objects. The setting is innovative and promising.

2. The proposed method is both simple and effective, as the authors employ an image customization technique to augment the dataset and address the challenge of limited training data.

3. Overall, the paper is well-structured and easy to read.

**Weaknesses:**

1. While the problem being studied is intriguing, the overall approach is relatively basic. The concepts of using image generation to augment the dataset and contrastive learning are not novel.

2. Additionally, incorporating image generation could significantly increase the cost of the method.

3. Moreover, in certain tasks, the method results in a decline in performance.

**Questions:**

1. Are there any real-world scenarios where the benefits of personalized representation outweigh the high training costs?

2. In certain challenging tasks, like distinguishing between dog faces of the same breed, it might be helpful to present more difficult or even failure cases to showcase the capabilities of the proposed method.

3. Why does the proposed method sometimes lead to a decline in final performance for certain tasks?

---

> ### Author Response · Authors · 2024-11-21
>
> Thank you for your helpful feedback and comments. We have carefully addressed your comments in detail below. Please let us know if you have any further questions.
>
> >***Novelty of the approach***
>
> Thank you for your feedback. We acknowledge that the individual concepts of synthetic data generation and contrastive learning are not novel. However, our work makes several unique contributions outside of the methodology components that we believe are valuable.
> * We empirically show that personalized representations (using synthetic data to handle lack of real data) significantly outperform pretrained representations, and that they are adaptable across different instance-level tasks.
> * We introduce new evaluation mechanisms for personalized representations in vision and present the PODS dataset, which we expect will be a useful benchmark for personalized representation, instance-level detection, and personalized generation research.
> * We highlight tradeoffs (computational cost v. downstream performance) of different approaches to personalized data generation, and the unique biases/limitations of different methods.
>
> These contributions offer new perspectives on personalized learning and synthetic augmentation in vision, with practical implications for future research. **We revised our introduction** to better clarify our contributions.
>
> >***Computational cost***
>
> Thank you for your insightful feedback on computational cost. We understand this is an important consideration and we address this question in **Sec. 2-3 of the Global Response**. We also add **updates to Section 5.2 of the paper** to study the tradeoffs between efficiency and performance. We compare methods involving image generation to those using solely real images, which is explained in **Sec. 1 of the Global Response**.
>
> >***Real-world scenarios where benefits outweigh training costs***
>
> You make a great point; it is important to consider real world cost-benefit tradeoffs. We discuss such scenarios in **Section 2 of our Global Response**.
>
> >***Challenging cases***
>
> Thank you for your suggestion -- **we add examples of challenging cases in Sec. D.2 of our revision (Figs. 16-17).**
>
> Given a target image, we identify hard negatives as test negatives with a *high* DINOv2 cosine similarity to the target image, and hard positives as test positives with a *low* cosine similarity. As you note, hard negatives are typically different dogs of the same breed; hard positives are often the same dog in very different contexts or poses.
>
> * Success cases: For hard positives DINOv2-P has higher cosine similarities than DINOv2, even under significant lighting, pose, and context shifts; this highlights its capabilities.
> * Failure cases: DINOv2-P also sometimes has higher cosine similarity for hard negatives – this could be because of noisy positives in the synthetic training dataset, causing the personalized representation to associate the target object with subtly incorrect features.
>
> >***Decline in performance for certain tasks***
>
> In the **Global Response Sec. 4** we explain some updates we have made to our main results. Personalization with DreamBooth now **outperforms non-personalization in 33/36 cases** in Table 1.
>
> However, you are right that in a few cases we see a decline in performance. We highlight some potential reasons:
> * State-of-the-art generators do not always achieve necessary levels of diversity/fidelity. For instance, with vanilla DreamBooth (Table 1), performance declines on classification for Dogs, but after incorporating masked training and filtering (Masked DreamBooth), performance increases (Table 2). We plot the **diversity/fidelity of these datasets v. performance in Fig. 12 of our revision**.
> * We fine-tune each backbone on the same data, however in practice different backbones may have unique strengths/weaknesses. For instance, performance on PODS/DF2 retrieval increases for DINO/CLIP, but declines for MAE.
>
> Overall, this shows there is room to improve for this novel problem formulation/setting beyond our proposed method.

---

> > ### Author Response · Authors · 2024-11-25
> >
> > Dear Reviewer FvPe,
> >
> > Thank you again for your detailed review! We have added difficult success and failure cases, addressed your comments regarding the novelty of our approach, clarified our contributions wrt computational cost, and explained declining performance for certain tasks in our responses and revision which we hope sufficiently address your questions/concerns. In the remaining short time, we would be very happy to alleviate any remaining concerns you still have about our paper.
> >
> > Thank you once again for dedicating your time and effort to reviewing our work and providing us with insightful suggestions!

---

> > > ### Author Response · Authors · 2024-11-30
> > > **A Kind Reminder**
> > >
> > > Dear Reviewer FvPe,
> > >
> > > Thank you again for reviewing our paper, and for the constructive feedback! We believe we have addressed all of the concerns/questions in your review, and updated our paper accordingly:
> > > - Discussed the **novelty of our contributions** in our initial response above and revised our **introduction to clarify**.
> > > - **Added runtimes** for methods to **Table 3** and **Sec. 5.2** of the paper (including breakdown by stage). We find that there exists a speed v. performance tradeoff, however performance gains are also achievable without T2I models (efficiently), using internet-available data and segmentation annotations.
> > > - Discussed **real-world scenarios** where performance benefits are critical and may outweigh the computational costs in Sec. 2 of the Global Response: Identifying personal items of blind/low-vision people, ecological applications such as animal reidentification, and medical imaging
> > > - Added examples and analysis of **challenging cases** (showcasing both capabilities and failure cases) in **Sec. D.2** and **Fig. 16-17** of the paper, which shows real-world generalizability of our personalized representations.
> > > - Discussed **why performance may decline in certain cases** in initial our response above: generators may not achieve needed levels of fidelity/diversity (using new analysis in Fig. 12), and different backbones may have unique strengths/weaknesses.
> > >
> > > For detailed discussion, please refer to our response above and the Global Response.
> > >
> > > As we approach the end of the discussion period, please let us know if you have further questions, or if there is anything else we can address for you to consider raising our score. Thank you again for your time and effort in reviewing our paper!

---

### Official Review · Reviewer_H1To · 2024-11-04

**Soundness:** 3
**Presentation:** 3
**Contribution:** 2
**Rating:** 6
**Confidence:** 4

**Summary:**

This paper investigates using personalized synthetic images, alongside a few real images, to develop personalized visual representations that enhance performance across multiple downstream tasks, including classification, retrieval, detection, and segmentation.

The authors propose a novel contrastive learning framework where a personalized representation is trained using only three real positive samples without any real negatives. To support the evaluation of this approach, they introduce the Personal Object Discrimination Suite (PODS) dataset, specifically crafted for testing under conditions such as pose variation and background distractors, along with reformulations of DeepFashion2 and DogFaceNet datasets for the same purpose. By leveraging generative models, particularly DreamBooth with masked training, they demonstrate the effectiveness of synthetic data for learning representations of specific instances. This personalized approach shows promising improvements over general-purpose pretrained representations.

**Strengths:**

The paper presents a creative combination of generative models and contrastive learning to address the challenge of instance-specific visual representation with minimal real data.

The experimental framework is thorough, covering classification, retrieval, detection, and segmentation tasks across three datasets (DeepFashion2, DogFaceNet, and the newly introduced PODS). The results consistently highlight the advantage of personalized representations over pre-trained ones, demonstrating the robustness of the approach. The author also shows that the method could be integrated into existing pipelines.

The paper is well-organized and has clear explanations. Multiple figures, including pipelines and qualitative results, aid in understanding. Writing is accessible without unnecessary complexity, making the proposed approach and findings straightforward.

**Weaknesses:**

1. It looks unnecessary to exclude real negatives in the proposed setting of personalized representation learning. Unlike real positives, real negatives might be easily obtained directly from open-source data. The paper relies on generated negatives produced by the generative model, which can be computationally costly. Alternatively, obtaining real negatives from readily available online sources might be a more efficient solution. Could the authors provide additional insights into why they excluded real negatives and whether this impacts performance? Are there any existing works exploring the setting of including real negatives?

2. Figure 1 is somewhat unclear and could be improved to more explicitly illustrate the flow and connection between real and synthetic data in the proposed framework. Additionally, it is not mentioned in the main texts. More explanation may also help.

3. The term "real-augmentation" is mentioned but not clearly defined in the context of this paper, which could lead to confusion regarding its role and implementation. By line 473, it seems that the real-aug is just using the real images without any augmentation to train the backbone models. If so, "real images" might be a better name than "real-aug."

4. The method of real augmentation on the PODS dataset shows degraded performance compared to pre-trained representations in retrieval, detection, and segmentation tasks, especially in retrieval (combining information from Table 1 and Table 2). However, this phenomenon does not appear in the other two datasets. This anomaly is not fully explored, and further insight into the underlying cause would help readers understand any limitations in generalizing the approach across different datasets.

**Questions:**

In addition to the questions in the "weaknesses" section:

1. In Table 1, are the results obtained with or without access to segmentation masks? This point is somewhat unclear—could the authors clarify if this information is noted elsewhere in the paper?

2. For tasks such as detection and segmentation, where the performance improvement was less pronounced, do the authors have plans to refine the pipeline for further gains in these areas? If so, could they share any specific strategies or potential modifications they are considering?

---

> ### Author Response · Authors · 2024-11-21
>
> Thank you for your constructive and insightful feedback. We address your comments in detail below. Please let us know if you have any further questions.
>
> >***Inclusion of real negatives***
>
> You make an important point about including real negatives. We initially excluded real negatives because we envisioned a setting in which a user can obtain personalized representations without any additional real data or upfront effort, besides taking a few positive images.
>
> We agree that in practice, real negatives may be available via open-source datasets. In our **Global Response, Sec. 1**, we explain updates to Sec 5.2 of the paper, including **results on allowing access to additional data**. We use real negatives from open-source datasets (ImageNet, LAION-RVS-Fashion [1]). Please refer to the Global Response for details/results, and let us know if any follow-up is needed.
>
> >***Revision of Figure 1***
>
> Thank you for your constructive feedback. **We have updated the figure** following your suggestions and added a reference to it in the introduction.
>
> >***Real-image baseline***
>
> We appreciate your point about the “real-aug” experiment. To clarify: in this experiment, we solely use the three real training images (i.e, $\mathcal{D}_R$) as positives. As suggested, **we have revised the name to  “real-image baseline”.**
>
> We always automatically apply on-the-fly classic augmentations (random horizontal flips, random rotations, and random resized crops) during training. These are applied to *all* positive/negative images, and can be considered a standard part of the contrastive training pipeline. **We have updated Sec. 4.2 of our paper** to clarify this. We do not report results on real-image or synthetic training without augmentations, as these are standard practice for vision pipelines.
>
> As you have noted, real-image training sometimes causes performance to decline. We suspect that this is due to the limited size of the real dataset (only 3 images). Even when applying classic augmentation, the dataset lacks diversity, and as shown in **Fig. 12 of our revision**, maintaining both fidelity and diversity are key to significant performance gains. We note that in our updated results, with a small training set, training on real-images improves on the pre-trained backbone in 8/12 cases.
>
> >***Access to segmentation masks***
>
> Thank you for pointing out the ambiguity. The distinction between our results in Sec. 5.1/Table 1 and Sec. 5.2/Table 2 is that Table 2 assumes access to additional resources including segmentation masks for training, while Table 1 only assumes access to the initial three real images. **We have added clarifications to Table 1 and Table 2.**
>
> We note, though, that for *downstream evaluation* of segmentation and detection, we always use segmentation masks, as we consider *evaluating/probing the learned representations* to be separate from actually *training the representations.*
>
> >***Dense task performance***
>
> To probe how well representations localize, we extract dense predictions directly from the patch embeddings using thresholding. This provides an unmodified assessment without introducing external factors such as pre-trained segmentation modules or additional post-processing.
>
> However, for weaker backbones like MAE and CLIP, while the object of interest is often identified as a high-confidence region, the confidence values are usually not distinct enough from the background, leading to very poor performance with simple thresholding, and thus marginal dense performance changes in Table 1. MAE features often require supervised finetuning to transfer to dense prediction tasks [2]. We point to DINOv2, which achieves good dense performance out-of-the-box, as a better assessment of how personalization affects dense performance. Future work could explore using personalized features to train segmentation heads (as in [3]); this would require generating segmentation masks for synthetic images.
>
> We also evaluate with state-of-the-art segmentation pipeline PerSAM in Table 9, which better reflects the practical use of these representations, and has more consistent performance gains.
>
> [1] Lepage, Simon et al. "LRVS-Fashion: Extending Visual Search with Referring Instructions." arXiv (2023).
>
> [2] He, Kaiming, et al. "Masked autoencoders are scalable vision learners." CVPR, 2022.
>
> [3] Oquab, Maxime, et al. "Dinov2: Learning robust visual features without supervision." TMLR, (2023).

---

> > ### Author Response · Authors · 2024-11-25
> >
> > Dear Reviewer H1To,
> >
> > Thank you again for your detailed review! We have added additional real negative baselines, updated our Figure 1, clarified ambiguities that you have kindly pointed out in our paper, and addressed performance inquiries around real augmentation and dense results in our responses and revision which we hope sufficiently address your questions/concerns. In the remaining short time, we would be very happy to alleviate any remaining concerns you still have about our paper.
> >
> > Thank you once again for dedicating your time and effort to reviewing our work and providing us with insightful suggestions!

---

> > > ### Author Response · Authors · 2024-11-30
> > > **A Kind Reminder**
> > >
> > > Dear Reviewer H1To,
> > >
> > > Thank you again for reviewing our paper, and for the constructive feedback! We believe we have addressed all of the concerns/questions in your review, and updated our paper accordingly:
> > > - **Results on real negatives** in **Table 2** and **Sec. 5.2** of the paper. We significantly extend our comparisons/baselines to study the tradeoffs between generated images and cheaper alternatives that leverage additional data/annotations. We add two new cases: 1) internet-available data and 2) scaling real positives, which allowed us to explore 5 additional baselines across different generative methods in our revision, including training with real negatives. .
> > > - **Updated Fig. 1** as requested and added reference to it in the Introduction.
> > > - Updated our **explanation of the real-image baseline** in **Sec. 4.2** of the paper; included clarification in our response above.
> > > - Discussed **why the real-image-only baseline can cause performance to decline** in our response above: in our new fidelity-diversity analysis (**Fig. 12**) we find that diversity is crucial to performance, and that synthetic datasets that closely mimic the real data perform worse; we hypothesize that this is why the even-more-limited real-only dataset can sometimes cause performance to decline.
> > > - Added clarification on **use of segmentation masks** to **Tables 1-2**, and in our initial response above.
> > > - Discussed **dense task performance** and strategies for refining the pipeline in our initial response above.
> > >
> > > For detailed discussion, please refer to our response above and the Global Response.
> > >
> > > As we approach the end of the discussion period, please let us know if you have further questions, or if there is anything else we can address for you to consider raising our score. Thank you again for your time and effort in reviewing our paper!

---

> > > > ### Comment · Reviewer_H1To · 2024-12-02
> > > >
> > > > Thank the authors for their efforts in conducting extra experiments and adding clarifications, especially those extra discussions. They have addressed my concerns. I am willing to maintain my original positive score.

---

### Official Review · Reviewer_CCxb · 2024-11-04

**Soundness:** 3
**Presentation:** 3
**Contribution:** 2
**Rating:** 6
**Confidence:** 3

**Summary:**

The paper aims to learn personalized representations given a small number of images of an object. This involves using T2I models to generate additional data followed by contrastive learning. A new dataset, PODS, was also introduced to allow for evaluations of these personalized representations under distribution shifts. Supportive results were shown for several tasks and datasets.

**Strengths:**

- The paper was well written and easy to follow.
- The overall idea of having a personalized vision backbone that can work well on several downstream tasks is interesting.
- The additional experiments on useful synthetic data was insightful.

**Weaknesses:**

- Additional baselines or comparisons.
    - For example, generating additional data with augmentations instead of using T2I models is a cheap baseline. If this is done in real-aug (Tab. 2) and it is comparable to the results of Tab. 1, why does it seem to perform worse than no personalization?
    - It may also be useful to compare against using real data only to get an upper bound of the method.
    - It would be interesting to see if, with more real images, cut/paste would outperform Masked DB. I.e., if we can tradeoff sampling more images for a faster runtime.
- The method seems to be computationally expensive as it involves several stages of finetuning and generation. It may be useful to include the runtimes of each method beside the results in e.g., Tab 2.
- The method learns a personalized representation for a single instance, a more realistic scenario probably involves several personalized instances e.g., one in each object category in PODS.

**Questions:**

- What is the runtime for the methods in Tab. 2? And what is the breakdown e.g., how much time take for each stage?
- Methods for synthetic data generation e.g., for supervised training, tend to include a filtering step to ensure that the generations are faithful to the prompt. Was any filtering needed and how was it done?
- What are some of the potential complications from extending the method to multiple instances?

---

> ### Author Response · Authors · 2024-11-21
>
> Thank you for your detailed review and valuable suggestions. We address your feedback below; please let us know if there are further comments or concerns.
>
> >***Additional baselines/comparisons***
>
> Thank you for your insightful comments regarding additional baselines and comparisons. **Please refer to Sec. 1 of our Global Response**, where we present results on the following additional comparisons:
> * Training with real negatives (rather than generated negatives) from the internet, i.e. from publically available vision datasets. This enables real-data-only baselines.
> * Training with an extended pool of real images, as an upper bound to performance, as well as Cut/Paste and Masked DreamBooth when trained/sampled from this extended pool.
>
> **Cut/Paste vs Masked DreamBooth trade-off:** When scaling up the real dataset to n=20, Cut/Paste begins to outperform DreamBooth (performance of 74% vs 70%, both outperform real-image only finetuning). However, sampling from the combined pool still performs best (76%), which suggests that T2I models contribute important additional information even with more real data. In domains where accuracy is key **(see Sec. 2 of global response)**, this method may be worth the extra compute.
>
> >***Computational cost and runtime***
>
> We agree that the computational cost is an important consideration. Please refer to **Sec. 2 of our Global Response**, where we address computational cost in detail. We have added the runtimes of our methods and comparisons in **Appendix Table 3**, with brief discussion in **Sec. 5.2** of the paper.
>
> >***Multi-object personalization***
>
> Thank you for your suggestion on multi-object personalization - it reflects a real-world use case of personalization, and an interesting extension to our work. We experimented with personalizing to multiple (2-5) objects on the PODS dataset by jointly training on triplets for multiple objects. With our current training pipeline, we can achieve strong improvements over pretrained models on global tasks (on par with single-object-personalized models). However, dense-task performance significantly decreases. Qualitatively, we found artifacts/noise in the patch-level similarity maps, causing distortions in the threshold detections/segmentations. Naiively training on triplets from multiple objects seems to negatively affect dense features, and require more careful exploration of the composition of triplets.
>
> >***Filtering synthetic data***
>
> For our results in Table 1 (i.e. personalization without access to additional data/annotations) we do not filter our synthetic datasets, however we do filter the Masked DreamBooth dataset (i.e., when segmentation masks are available).
>
> You are correct that many works, such as [1], apply a filtering step. In our preliminary experiments we found that filtering without segmentation masks of the target object did not meaningfully improve performance. This is likely because we have a very small reference train set (3 images) so filtering based on global embeddings is affected by spurious signals, such as keeping images with the same backgrounds as training images. This negatively affects diversity, outweighting improvements in fidelity.
>
> When segmentation masks are available, we are able to decouple the object and background in our training images. **We explain our filtering procedure in Sec. A.3.2.**
>
> In **Fig. 12 of our revision (Appendix Sec. C.4)**, we plot the relationship between fidelity, diversity, and resultant performance for DreamBooth datasets (with/without filtering and LLM captions) and Cut/Paste. While non-filtered DreamBooth datasets lie at the extremes (high-diversity, low-fidelity or vice versa), the filtered dataset achieves a better balance of the two, and thus higher performance.
>
> >***Performance on real-image personalization***
>
> As you have noted, training on real-images only sometimes causes performance to decline. We suspect that this is due to the limited size of the real dataset (only 3 images). Even when applying classic augmentation, the dataset lacks diversity, and as shown in **Fig. 12 of our revision**, maintaining both fidelity and diversity are key to significant performance gains. We note that in our updated results, with a small training set, training on real-images improves on the pre-trained backbone in 8/12 cases.
>
> [1] He et al. "Is synthetic data from generative models ready for image recognition?." ICLR 2023.
>
> [2] Zhang et al. “Personalize segment anything model with one shot”. arXiv, 2023.
>
> [3] Fu et al. “DreamSim: Learning new dimensions of human visual similarity using synthetic data”. NeurIPS, 2023.

---

> > ### Author Response · Authors · 2024-11-25
> >
> > Dear Reviewer CCxb,
> >
> > Thank you again for your detailed review! We have added additional baselines, addressed your questions wrt computational cost, and explored multi-object personalization in our responses and revision which we hope sufficiently address your questions/concerns. In the remaining short time, we would be very happy to alleviate any remaining concerns you still have about our paper.
> >
> > Thank you once again for dedicating your time and effort to reviewing our work and providing us with insightful suggestions!

---

> > > ### Author Response · Authors · 2024-11-30
> > > **A Kind Reminder**
> > >
> > > Dear Reviewer CCxb,
> > >
> > > Thank you again for reviewing our paper, and for the constructive feedback! We believe we have addressed all of the concerns/questions in your review, and updated our paper accordingly. To summarize:
> > > - **Added requested extra baselines** to **Table 2** and **Sec. 5.2** of the paper. We significantly extend our comparisons/baselines to study the tradeoffs between generated images and cheaper alternatives that leverage additional data/annotations. We add two new cases: 1) internet-available data and 2) scaling real positives, which allowed us to explore 5 additional baselines across different generative methods in our revision.
> > > - **Added requested runtimes** for methods to **Table 3** and **Sec. 5.2** of the paper (including breakdown by stage). We find that there exists a speed v. performance tradeoff, however performance gains are also achievable without T2I models (efficiently), using internet-available data and segmentation annotations.
> > > - Clarification of **filtering procedures** in our response above and in Sec. A.3.2. We do apply filtering in the Masked DreamBooth method, and find that it increases performance.
> > > - Added analysis of **how fidelity/diversity affects performance (esp after filtering)** in **Fig. 12** in the paper. We show that filtering improves the balance of fidelity/diversity.
> > > - Discussed preliminary results from **extending to multiple instances**, and complications that arise in dense tasks, in our initial response above.
> > > - Discussed **why the real-image-only baseline can cause performance to decline** in our initial response above: in our fidelity-diversity analysis we find that diversity is crucial to performance, and that synthetic datasets that closely mimic the real data perform worse; we hypothesize that this is why the even-more-limited real-only dataset can sometimes cause performance to decline.
> > >
> > > For detailed discussion, please refer to our response above and the Global Response.
> > > As we approach the end of the discussion period, please let us know if you have further questions, or if there is anything else we can address for you to consider raising our score. Thank you again for your time and effort in reviewing our paper!

---

### Author Response · Authors · 2024-11-21
**Global Response**

We thank all of the reviewers for their insightful and helpful feedback. We are glad that they found:
* The paper was well-written [CCxb,H1To,FvPe,eQii]
* The experiments are insightful/thorough [CCxb,H1To]
* The paper studies an interesting, novel problem [CCxb,FvPe,2SHx,eQii]

We highlight key revisions (and answer individual questions in reviewer-specific responses). **Updates are blue in our revised PDF**.

**1. ADDITIONAL COMPARISONS**

**We extend our comparisons/baselines to study the tradeoffs** between training on T2I-generated images – which is computationally costly, but does not require further data/annotation – and cheap alternatives that leverage additional data.

Segmentation masks enable Cut/Paste (CP) with generated backgrounds, and Masked DreamBooth (Masked DB). In our revision we add two new cases:

**1) Internet-available data:** We use open-source vision datasets as a source of non-customized real data. These can be used as *real negatives* for contrastive learning and *real-backgrounds* for CP. This enables the following comparisons:
* A real-only baseline: (real images from $\mathcal{D}_R$ as positives, real negatives).
* CP with real backgrounds and real negatives
* Generated (Masked DB) positives with real negatives.

**Results are in Table 2 of our revision.** We find:
* Methods using open-source real images perform similarly to using generated negatives or CP backgrounds.
* Methods using synthetic positives outperform the real-only baseline; Masked DB + CP perform best.
* Strong performance is possible with a compute-efficient alternative (CP with real backgrounds and real negatives).

**2) Scaling real positives:** A user may manually expand $\mathcal{D}_R$ by collecting extra real positives. **In Fig. 6 of our revision we scale $\mathcal{D}_R$** and evaluate training with:
* Real images from the larger $\mathcal{D}_R$
* The best-performing synthetic method (combined CP and Masked DB) using the larger $\mathcal{D}_R$.

We manually collect extra real training images for 25 instances of PODS (5 per category), and show results for real-image training on $|\mathcal{D}_R| \in \\\{3, 5, 10, 15, 20\\\}$. Due to time constraints, we show synthetic results at $|\mathcal{D}_R| \in \\\{3, 20\\\}$. We cannot replicate this experiment on DF2/Dogs due to the limited positives per class.

We observe:
* Real-image training saturates at n=15. While our extra images include new backgrounds/poses/lighting, diversity can still be limited, which we show is key to accuracy gains via **diversity/fidelity analysis in Figure 12**. We aimed to mimic users photographing their objects; thus our training images mainly are single-object images in indoor environments. Better scaling may be possible with significant extra diversity (though this requires intentional, potentially labor-intensive, data collection).
* Even after real images saturate, synthetic data improves performance. This is likely because the generator (Masked DB + CP) can more readily achieve per-image diversity.

**2. COMPUTATIONAL COST/RUNTIME**

**We report runtimes of each synthetic data method in Table 3** and find a speed v. performance tradeoff. Masked DB + CP performs best, but is also expensive; the real-image baseline is extremely cheap but performs worst. However, performance gains are also achievable without T2I models. CP with real-backgrounds/real-negatives – the fastest method in Table 3 – outperforms the real-image baseline. Thus our results do not rely solely on T2I models, although these help to achieve the best results.

There are also domains where high performance/robustness is critical, and downstream accuracy gains outweigh computational cost:
* Personalized vision systems are useful for identifying personal items of blind or low-vision people [1].
* In ecology, re-identifying specific animals is an open problem; real data collection is often extremely expensive/difficult.
* Personalization is becoming increasingly useful in medical imaging.

**3. CONTRIBUTION w.r.t COST**

We emphasize that our main contribution – a framework for personalized representations, and exploration of performance benefits – is agnostic to the choice of generator. In future, especially considering the recent pace of progress, generation and fine-tuning will only become more efficient. While we focus on elucidating the *benefits* of personalized representations, we are optimistic that in the future, more efficient techniques can be easily plugged in to further reduce the *cost*. **We have revised our introduction to clarify our contributions.**

**4. CORRECTED RESULTS**

We found a small bug related to image loading, and have updated results to reflect the correction. There is very little change, however we highlight that in Table 1, **personalized backbones outperform pretrained backbones in 33/36 cases.**

---

> ### Author Response · Authors · 2024-11-21
>
> **References**
>
> [1] Morrison, et al. “Understanding Personalized Accessibility through Teachable AI: Designing and Evaluating Find My Things for People who are Blind or Low Vision.” SIGACCESS Conference on Computers and Accessibility, 2023.

---

> > ### Author Response · Authors · 2024-11-25
> > **Highlighting other revisions**
> >
> > Dear Reviewers,
> >
> > Thank you again for reviewing our paper! In addition to the main revisions, we would also like to highlight some of the smaller revisions that were not mentioned in our global response:
> > * Results on generalization:
> >   * Qualitative examples of challenging real-world cases that we added in **Section D.2 (Fig. 16-17) of our revision** (comparing personalized and pretrained models).
> >   * **We updated Fig. 9** to compare how synthetic-learned representations and real-learned representations generalize to real-world distribution shifts, and we find that synthetic-learned representations significantly out-perform real-learned representations in OOD test-sets.
> >   * **We added Fig. 10-11**: Examples of the Dogs/DF2 test sets, demonstrating that they test real-world scenarios and distribution shifts
> > * Fidelity-diversity analysis:
> >   * **Added Section C.4 (Fig 12)** we plot and discuss the diversity v. fidelity (colored by performance) of different synthetic datasets, where we find that balancing the two characteristics is important for optimal downstream performance.

---

### Meta-Review · Area_Chair_fxsY · 2024-12-19

**Metareview:**

This paper aims to address the challenge of using synthetic data to enhance personalized visual representation learning for fine-grained and data-scarce tasks. The authors propose a contrastive learning framework leveraging text-to-image generative models, like DreamBooth, to create synthetic images for instance-specific representation learning.

This paper received mixed ratings before the rebuttal period. After reading the author's responses, three of the four reviewers upgraded their rating to the positive side or decided to keep their positive ratings. The only one who didn't change their score is Reviewer 2ShX. However, they didn't actively participate in the discussions. Considering all of the above, it is decided to accept the paper. The authors are required to include the discussions in their final version.

**Additional Comments On Reviewer Discussion:**

This paper received mixed ratings before the rebuttal period. After reading the author's responses, three of the four reviewers upgraded their rating to the positive side or decided to keep their positive ratings. The only one who didn't change their score is Reviewer 2ShX. However, they didn't actively participate in the discussions. Considering all of the above, it is decided to accept the paper. The authors are required to include the discussions in their final version.

---

### Decision · Program_Chairs · 2025-01-22

Accept (Poster)